# Interleukin 22 disrupts pancreatic function in newborn mice expressing IL-23

Lili Chen [1,8], Valentina Strohmeier[1,2,8], Zhengxiang He [1,8], Madhura Deshpande[1], Jovani Catalan-Dibene [1], Scott K. Durum[3], Thomas M. Moran[4,5], Thomas Kraus[5], Huabao Xiong[1], Jeremiah J. Faith [1,6], Chhinder P. Sodhi[7], David J. Hackam [7], Sergio A. Lira [1]* & Glaucia C. Furtado [1]*

Neonatal inflammatory diseases are associated with severe morbidity, but the inflammatory factors underlying them and their potential effector mechanisms are poorly defined. Here we show that necrotizing enterocolitis in neonate mice is accompanied by elevation of IL-23 and IL-22 and decreased production of pancreatic enzymes. These phenotypes are mirrored in neonate mice overexpressing IL-23 in CX3CR1[+] myeloid cells or in keratinocytes. The mice fail to grow and die prematurely, displaying systemic inflammation, nutrient malabsorption and decreased expression of intestinal and pancreatic genes mediating digestion and absorption of carbohydrates, proteins, and lipids. Germ-free environment improves, and genetic ablation of IL-22 restores normal growth in mice overexpressing IL-23. Mechanistically, IL-22 acts directly at the level of pancreatic acinar cells to decrease expression of the pancreas associated transcription factor 1a (PTF1a). These results show that augmented production of IL-23 and IL-22 in early life has a negative impact on pancreatic enzyme secretion and food absorption.

[1] Precision Immunology Institute, Icahn School of Medicine at Mount Sinai, New York, NY 10029, USA. [2] Faculty of Biology, University of Freiburg, Schaenzlestrasse 1, 79104 Freiburg, Germany. [3] Center for Cancer Research, National Cancer Institute, Frederick, MD 21702, USA. [4] Department of Microbiology, Icahn School of Medicine at Mount Sinai, New York, NY 10029, USA. [5] Center for Therapeutic Antibody Development, Icahn School of Medicine at Mount Sinai, New York, NY 10029, USA. [6] Institute for Genomics and Multiscale Biology, Icahn School of Medicine at Mount Sinai, New York, NY 10029, USA. [7] Division of General Pediatric Surgery, Johns Hopkins University and Bloomberg Children's Center, Johns Hopkins Hospital, Baltimore, MD 21287, USA. [8] These authors contributed equally: Lili Chen, Valentina Strohmeier, Zhengxiang He. *email: sergio.lira@mssm.edu; glaucia.furtado@mssm.edu

The gut of a newborn mouse is immature, resembling that of a premature infant[1,2]. It lacks cells specialized in antimicrobial defense, such as Paneth cells[3], contains few T and B cells, and it is populated mostly by myeloid immune cells[2,4]. The lymphoid compartment at this stage is represented mostly by innate lymphoid cells (ILCs) that can respond to bacteria, and are important for the formation of structures such as Peyer's patches (PPs) and mesenteric lymph nodes[5]. Maturation of the intestinal barrier and the immune system appears to be dependent on colonization by commensal bacteria[6,7]. Colonization of the neonatal intestine starts at birth, with the newly acquired bacteria coming from the vaginal canal or skin, and triggering a complex interaction with cells of the immune system[6]. This process is further influenced by immunoregulatory factors present in the maternal milk[8].

Precise and balanced responses are performed by the cells of the innate immune system in the intestine of newborns, but in a number of occasions, the responses are not balanced, and significant immunopathology arises. One of the important regulators of myeloid and ILC biology is the cytokine interleukin-23 (IL-23)[9,10]. IL-23 has been implicated in several inflammatory diseases in adults, but its role in neonates remains only partially characterized. Deficit in IL-23 signaling is not associated with a major deficit in the immune function in newborn mice, but overexpression of IL-23 in the neonatal gut promotes severe intestinal inflammation, intestinal bleeding, and perinatal death[11].

To further understand the pathological consequences of IL-23 expression in early life, we generated two novel mouse strains in which IL-23 was overexpressed in CX3CR1-positive myeloid cells and keratinocytes, respectively. We show here that increased expression of IL-23 leads to stunted growth and premature death in both mouse strains. The phenotype is mediated by the IL-23 downstream cytokine IL-22, whose levels are increased in circulation. Increased levels of IL-22 in circulation of newborn mice affect the expression of genes encoding antimicrobial peptides and digestive enzymes produced by the pancreas and intestine, resulting in a malabsorptive condition. Deletion of IL-22 in mice expressing IL-23 reverses these changes and prevents early death. Our results reveal an unsuspected role for IL-23-induced IL-22 in controlling pancreatic enzyme secretion and food absorption in early life.

## Results

**Early lethality in mice expressing IL-23 in CX3CR1+ cells.** Animals expressing IL-23 from the villin promoter (V23 mice) die at birth[11]. The cause of death appears related to intestinal bleeding originating from the small intestine (SI). To further study the biology of IL-23 in neonates, we engineered mice in which expression of IL-23 was targeted constitutively to CX3CR1+ cells, which are the cells that predominantly express IL-23 in the intestine. This was accomplished by intercrossing mice containing a IL-23 cassette preceded by a floxed STOP signal in the ROSA26 locus (R23 mice)[12], with mice containing a cre-recombinase gene inserted into the CX3CR1 locus (CX3CR1-cre mice)[13] (Fig. 1a). We refer to these animals as CXR23 mice (Fig. 1a). Expression of the IL-23 subunits p19 and p40 was detected in the intestine of the CXR23, but not control mice (Fig. 1b, c). CXR23 mice were normal at birth, but ~50% of the pups died within the first 48 h of life, with the remaining pups perishing before day 8 (Fig. 1d). CXR23 mice had a normal body weight at birth, but did not grow after birth (Fig. 1e, f). Necropsy of the newborn mice showed the presence of blood within the SI. The histological analyses of the CXR23 intestine showed that the bleeding originated from disrupted villi and from cellular aggregates that resembled PP anlagen (Fig. 1g). The lesions were present in the SI (duodenum, jejunum) but not in the colon (Supplementary Fig. 1a). We next determined by flow cytometry, the number and localization of CX3CR1+ cells in the intestine of mice at birth. We found that the number of CX3CR1+ cells in the SI of wild-type (WT) mice were higher than that found in the large intestine (Fig. 1h and Supplementary Fig. 2a). The number of CX3CR1+cells in the SI of CXR23 mice was 2-fold higher than that found in the SI of controls (Fig. 1i and Supplementary Fig. 2a). The cellular aggregates in the CXR23 mice were rich in neutrophils that disrupted the overlaying epithelium (Fig. 1g), and in IL-22+ cells (Fig. 1j), suggesting a role for these cells in pathology. Of note, the number of ILC3, potentially capable of producing IL-22 and IL-17 upon stimulation with IL-23[11], was markedly increased in the SI of CXR23 mice compared to controls (Fig. 1k and Supplementary Fig. 2b). No abnormalities were found in other organs (kidney, heart, lung, and brain) by conventional histological analyses (Supplementary Fig. 1a). Together, these findings confirm our previous observations that IL-23 expression in the murine gut results in early lethality[11].

**Germ-free CXR23 mice have increased lifespan.** Previous work from our lab suggested that expression of IL-23 could modify intestinal permeability and facilitate bacterial translocation during the immediate neonatal period[11]. To investigate whether bacteria contributed to the phenotype of early lethality, we generated CXR23 mice in germ-free (GF) conditions (referred to as CXR23 GF mice). As indicated before, ~50% of CXR23 SPF mice die within the first 48 h after birth. More than 95% of the CXR23 GF mice were alive at this point, and survived up to 30 days of age (Supplementary Fig. 3a). No bleeding was observed in the intestine of CXR23 GF mice at birth or later (Supplementary Fig. 3b). Neutrophils were present in the SI, but did not disrupt the epithelium (Supplementary Fig. 3c). Together, the results suggest that the newly acquired microbiota contributes to the development of the intestinal bleeding phenotype observed in CXR23 SPF neonates.

**Early lethality in mice expressing IL-23 in the skin.** To further investigate the factors contributing to early lethality and stunted body growth elicited by IL-23 expression, we engineered another strain in which expression of IL-23 was directed to the keratinocytes, by intercrossing the R23 mice[12] with mice carrying a cre-recombinase driven by the K14 promoter (Fig. 2a). These animals, which we refer to as KS23 mice, expressed higher levels of IL-23 in the skin than control (WT) littermates (Fig. 2b, c) and expressed higher levels of IL-23 in the skin than in the other organs (tongue, esophagus, and gut) (Fig. 2b, c). The KS23 mice had normal body weight at birth (Fig. 2d). By day 5 KSR23 mice had milk in the stomach, but were smaller than their control littermates (Fig. 2d, e). Similar to CXR23 mice, KSR23 mice died prematurely (before 15 days of age) (Fig. 2f), but displayed no signs of disease in the skin, intestine, or in other organs examined (Supplementary Fig. 1b). A notable finding in the animals was the presence of diarrhea, with yellow stools (Fig. 2g), which suggested food malabsorption. Conditions that lead to malabsorption are associated with increased fecal fat content, commonly known as steatorrhea. To determine if KSR23 mice developed steatorrhea, we measured triglycerides (TGs) in the stools and in the serum (Fig. 2h, i). At day 5, KSR23 mice had increased fecal TG (Fig. 2h) and decreased TG levels in the serum (Fig. 2i) when compared to WT littermate controls. Together, these results indicate that constitutive expression of IL-23 in keratinocytes results in fat malabsorption, growth retardation, and early death.

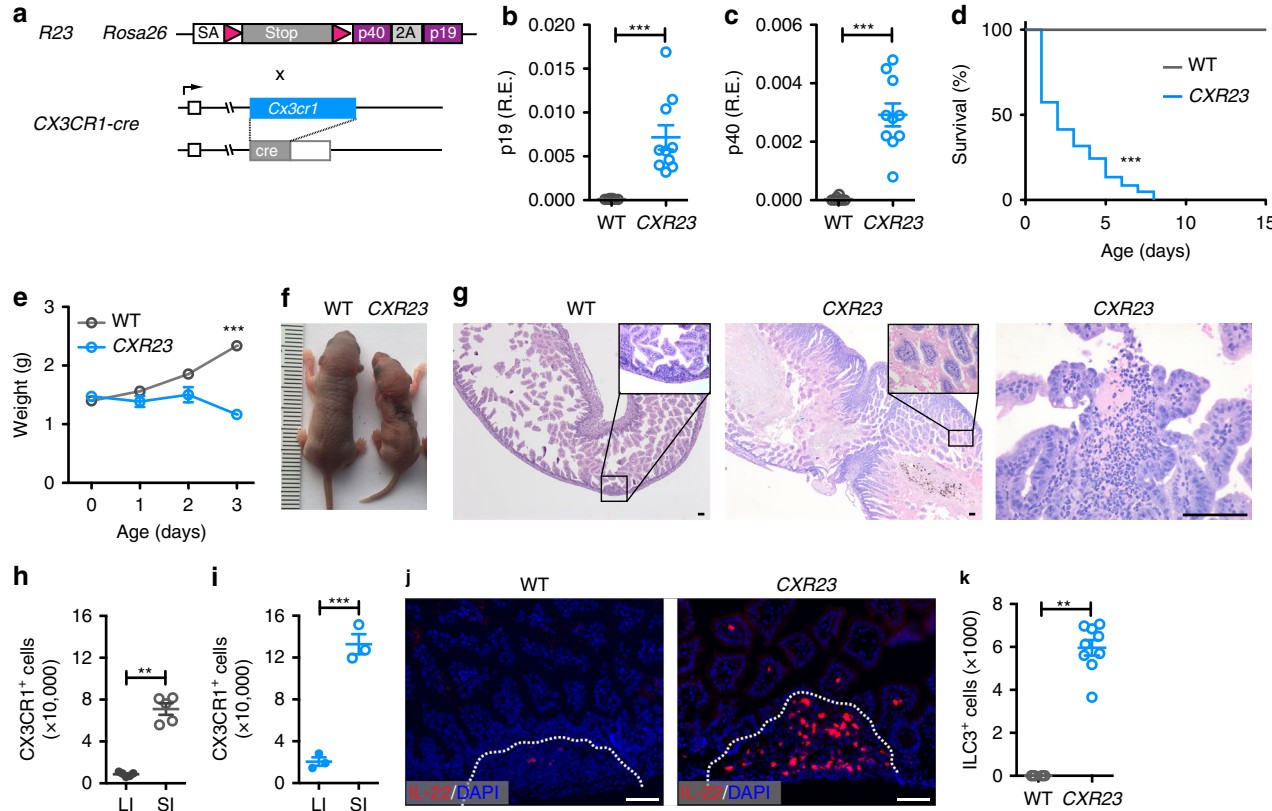

**Fig. 1** Constitutive expression of IL-23 in CX3CR1+ cells results in early lethality. **a** Scheme for generation of *CXR23* mice. *R23* mice containing a knock-in of IL-23p19 and p40 in the ROSA26 locus were crossed to *CX3CR1-cre* mice to generate *CXR23* mice. **b**, **c** Relative expression of *p19* (**b**) and *p40* (**c**) mRNA in the intestine of WT and *CXR23* mice at postnatal day 1 (P1) (*n* = 10 mice/group). **d** Survival curves of WT and *CXR23* mice (WT, *n* = 35, *CXR23*, *n* = 87). **e** Body weight of WT and *CXR23* mice (*n* = 11–24 mice/group). **f** Representative picture of WT and *CXR23* mice at P3. **g** Representative H&E-stained section of the small intestine of WT and *CXR23* mice at P1. Inset shows the presence of red blood cells in the intestine of *CXR23* mice. Representative picture of an erosive lesion in the small intestine of *CXR23* mice at P1 (right panel). Scale bars = 50 μm. **h**, **i** Flow cytometric analysis of CX3CR1+ cells in the large (LI) and small (SI) intestines of wild-type (**h**) and *CXR23* (**i**) mice at postnatal day 1 (*n* = 3–5 mice/group). **j** Immunostaining of the small intestine of *IL-22tdTomato* (WT) and *CXR23/IL-22tdTomato* (*CXR23*) mice at P1 with anti-tdTomato antibody. Notice the accumulation of IL-22-positive cells in the intestine of *CXR23* mice with erosive lesions (right panel). Scale bars = 50 μm. **k** Total number of group 3 innate lymphoid cells (ILC3+) cells in the small intestine of *CXR23* mice at P1. ILC3+ cells were gated on CD45+Lin−Thy1+Sca-1hi (*n* = 8 mice/group). Data are shown as mean ± SEM. Statistical analysis using nonparametric Mann–Whitney test for **b**, **c**, **h**, **i**, and **k**. Statistical analysis using a log-rank test for **d**. Statistical analysis using ANOVA with Bonferroni post hoc test for **e**. **\*\***p < 0.01 and **\*\*\***p < 0.001. Source data are provided as a Source Data file

**Increased serum levels of cytokines in mice expressing IL-23.** To start evaluating the mechanisms leading to stunted growth and early lethality associated with IL-23 expression, we assayed the levels of IL-23 and other cytokines in the serum of *CXR23* and *KSR23* mice. Levels of IL-23 were elevated in the serum of *CXCR3* and *KSR23* mice (~7- and 2.5-fold over controls, respectively) (Supplementary Fig. 4a and 4b). Serum levels of IL-22, IL-17, interferon-γ (IFNγ), IL-1β, but not granulocyte–macrophage colony-stimulating factor were significantly elevated in the serum of *CXR23* mice (Supplementary Fig. 4a). Levels of IL-22, IFNγ, and IL-1β were significantly elevated in the serum of *KSR23* mice (Supplementary Fig. 4b). In general, the levels of cytokines in serum correlated with the levels of IL-23, with the changes being more marked in the *CXR23* mice. The most upregulated cytokine in the serum of *CXR23* and *KSR23* mice was IL-22, whose overall levels reached 10 and 4 ng/ml, respectively (Supplementary Fig. 4a, b). The markedly elevated levels of cytokines in serum suggested a possible role for these molecules in pathogenesis.

**Genetic ablation of IL-22 reduces lethality.** To test if IL-22 had a role in the pathogenesis elicited by IL-23 in vivo, we intercrossed *CXR23* and *KS23* mice to IL-deficient (*IL-22−/−*) mice generated

in our laboratory (Supplementary Figs. 5 and 2c). IL-22-deficient *CXR23* (*CXR23/IL-22−/−*) and *KSR23* (*KSR23/IL-22−/−*) mice had extended lifespans (Fig. 3a, d) and normal growth (Fig. 3b, e), suggesting that IL-22 had a major role in the stunted growth and lethality phenotypes exhibited by these animals. Of note, none of the IL-22-deficient *CXR23* mice examined had intestinal bleeding (Fig. 3c). To further document a role for IL-22 in pathogenesis, we examined animals that still expressed IL-23 and IL-22, but lacked one of the chains of the IL-22R (IL-10R2). These mice are referred to as *CXR23/IL-10R2−/−* mice. Similar to what was observed above, animals in which IL-22 signaling was inactivated survived longer (Fig. 3a), and had normal body weight at P3 (Fig. 3b). Together, these results indicate that IL-22 is the main cytokine driving neonatal pathology in mice expressing IL-23 at birth.

**Dysregulated gene expression in intestine and pancreas.** IL-23 and other cytokines were detected at higher levels in the serum of *CXR23* and *KSR23* mice than WT mice (Supplementary Fig. 4). We reasoned that these changes could have affected the growth of the newborn mice. Body growth in newborns is dependent on their ability to ingest, process, and absorb nutrients. *CXR23* and

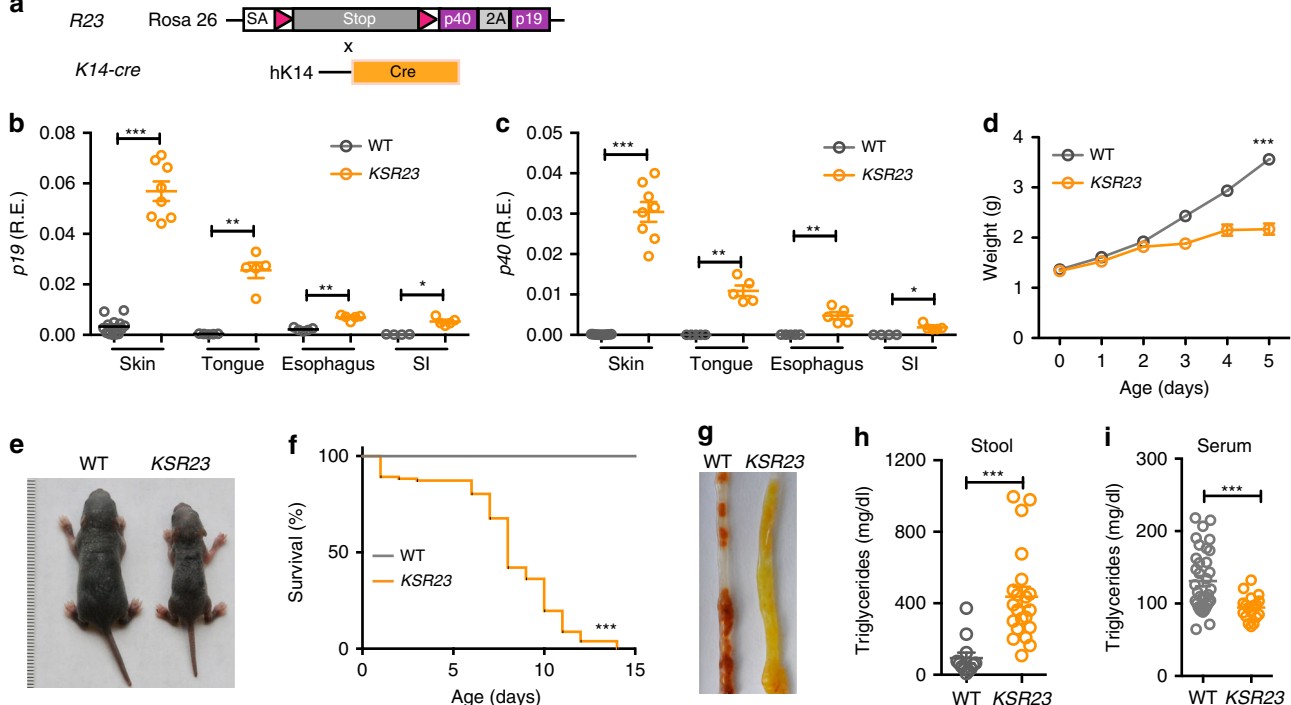

**Fig. 2** Constitutive expression of IL-23 in K14+ cells results in early lethality. **a** Scheme for generation of *KSR23* mice. *KSR23* mice were generated by crossing *R23* mice with mice expressing cre-recombinase from the K14 promoter (*K14-cre* mice). **b, c** Relative expression of *p19* (**b**) and *p40* (**c**) mRNA in several organs of WT and *KSR23* mice at P7 (*n* = 4–13 mice/group). **d** Body weight of WT and *KSR23* mice during the first week of life (*n* = 14–26 mice/group). **e** Representative picture of WT and *KSR23* mice at P5. **f** Survival curves of WT and *KSR23* mice (WT, *n* = 54; *KSR23*, *n* = 105). **g** Representative picture of the large intestine of WT and *KSR23* mice at day 5. Notice the presence of soft yellow stool in the colon of *KSR23* mice. **h** Quantification of triglycerides in the stool of WT (*n* = 12) and *KSR23* (*n* = 21) mice at P5. **i** Quantification of triglycerides in the serum of WT (*n* = 35) and *KSR23* (*n* = 18) mice at P5. Data are shown as mean ± SEM. Statistical analysis using one-way ANOVA for **b**, **c**. Statistical analysis using ANOVA with Bonferroni post hoc test for **d**. Statistical analysis using a log-rank test for **f**. Statistical analysis using nonparametric Mann–Whitney test for **h**, **i**. *$p < 0.05$, **$p < 0.01$, and ***$p < 0.001$. Source data are provided as a Source Data file

*KSR23* mice had milk in the stomach, suggesting that food ingestion was not affected. To investigate if elevated cytokines could affect body growth, we examined the transcriptome of the pancreas and the intestine, organs that are critical for the processing and absorption of nutrients. To do so, we extracted RNA from the pancreas and jejunum of 5-day-old mice expressing IL-23 in the skin (*KSR23* mice) and performed RNA-sequencing (RNASeq).

As shown in Fig. 4a, b, the transcriptome of the pancreas of *KSR23* mice at P5 differed significantly from that of their control littermates. The KEGG pathway analysis showed that several genes involved in pancreatic secretion were downregulated in the pancreas of *KSR23* mice (Fig. 4c). The pancreatic secretion pathway includes genes encoding several pancreatic enzymes involved in food digestion and genes involved in the secretory pathway (Fig. 4c). Among the genes downregulated in the *KSR23* mice were those encoding pancreatic enzymes, including pancreatic amylases (amylase 2a, *Amy2a*; amylase 2b, *Amy2b*), lipase (*Pnlip*), elastase (elastase 1, *Cela1*; elastase 2a, *Cela2a*), and proteases (carboxypeptidase 1, *Cpa1*; carboxypeptidase 2, *Cpa2*; kallikrein 1-related peptidase b5, *Klk1b5*; chymotrypsin-like protease *Ctrl*; chymotrypsin C, *Ctrc*) (Fig. 4d). Other genes involved in pancreatic acinar secretion were also downregulated, including syncollin (*Sycn*), a protein that regulates the fusion of zymogen granules[14]. Aquaporin 12 (*AQP12*), an acinar-specific water channel that controls secretion of pancreatic fluid[15], was also downmodulated in the pancreas of *KSR23* mice (Fig. 4e).

Some of these changes were confirmed by quantitative PCR (qPCR) (Fig. 4f).

The transcription regulatory gene *Ptf1a* controls acinar cell secretory protein processing and packaging[16]. Inactivation of *Ptf1a* in adult acinar cells results in decreased acinar cell identity and this coincides with the appearance of the ductal cell markers Sox9 and keratin19 (KRT19)[16,17]. We observed decreased expression of *Ptf1a* (Fig. 5a) and increased expression of *Sox9* (Fig. 5b) in the pancreas of *KSR23* mice. These results suggest that acinar cells in *KSR23* mice may acquire ductal cell phenotype. To determine if this was indeed the case, we analyzed the expression of KRT19 in the pancreas of WT and *KSR23* mice by immunostaining (Fig. 5c). Increased number of KRT19+ cells was observed in the pancreas of *KSR23* mice when compared to WT mice (Fig. 5c). KRT19 expression mainly localized to the large interlobular ducts in WT pancreas (Fig. 5c, arrow). However, in *KSR23* mice, KRT19 also marked a population of acinar cells (Fig. 5c, arrow). These results further indicate that acinar cell function and identity are disturbed in the pancreas of *KSR23* mice.

Trypsinogens, chymotrypsinogens, lipases, elastases, and proteases are produced by the acinar cells and secreted via the pancreatic ducts into the SI. At this site, some of these precursors are converted into active forms, by enzymes produced by the SI, leading to further breakdown of food. Peptides, amino acids, fatty acids, glycerol, and glucose reach the bloodstream via transporters present in the brush border membrane of intestinal cells. To ask if increased circulating levels of IL-23 and other cytokines

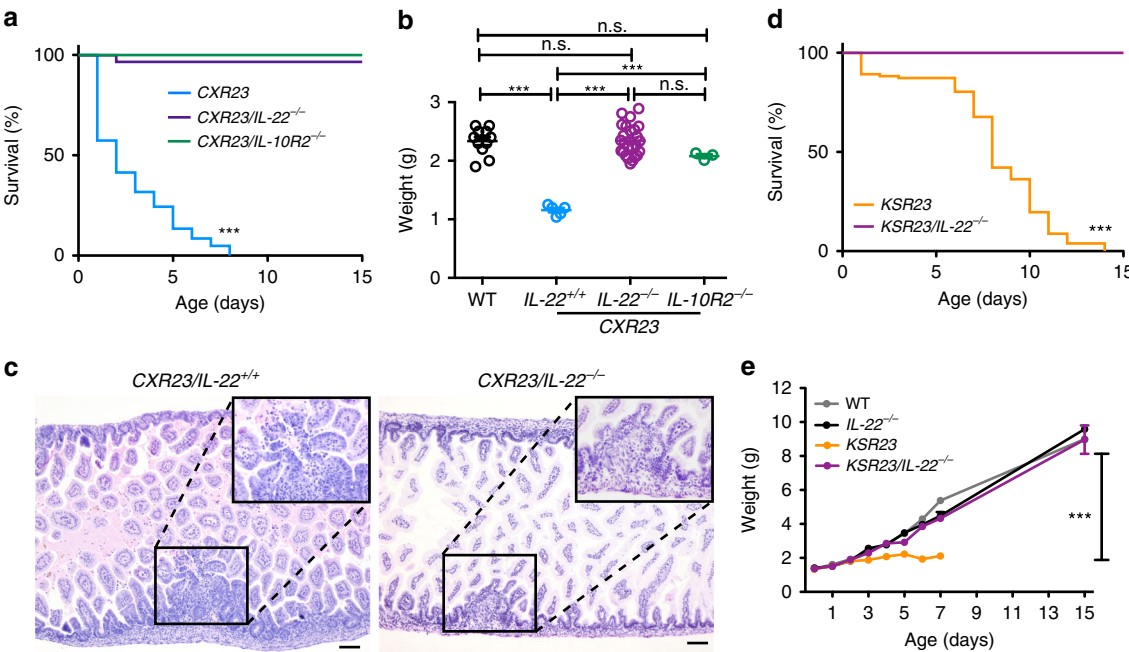

**Fig. 3** Genetic ablation of IL-22 or IL-10R2 reduces lethality and restores normal body growth in neonates expressing IL-23. **a** Survival curves of *CXR23*, *CXR23/IL-22⁻/⁻*, and *CXR23/IL-10R2⁻/⁻* mice (*CXR23*, n = 87, *CXR23/IL-22⁻/⁻*, n = 28, *CXR23/IL-10R2⁻/⁻*, n = 12). **b** Body weight of WT, *CXR23*, *CXR23/ IL-22⁻/⁻*, and *CXR23/IL-10R2⁻/⁻* at P3 (n = 3–25 mice/group). **c** Representative H&E-stained sections of the small intestine of *CXR23/IL-22⁺/⁺* and *CXR23/IL-22⁻/⁻* mice. Inset shows higher magnification of areas with erosive lesions and lymphoid aggregates in the intestine of *CXR23/IL-22⁺/⁺* and *CXR23/ IL-22⁻/⁻* mice, respectively. Scale bars = 50 μm. **d** Survival curves of *KSR23* and *KSR23/IL-22⁻/⁻* mice (*KSR23*, n = 108; *KSR23/IL-22⁻/⁻*, n = 25). **e** Weight curves of WT, *IL-22⁻/⁻*, *KSR23*, and *KSR23/IL-22⁻/⁻* mice over time (n = 5–33 mice/group). Data are shown as mean ± SEM. Statistical analysis using a log-rank test for **a**, **d**. Statistical analysis using one-way ANOVA for **b**. Statistical analysis using ANOVA with Bonferroni post hoc test for **e**. ⁿ·ˢ·p > 0.05 and ***p < 0.001. Source data are provided as a Source Data file

were also associated with marked changes in the expression of genes in the intestine, we analyzed extracted RNA from the jejunum of *KSR23* and control mice and performed RNASeq. The overall pattern of gene expression differed significantly between *KSR23* and control mice (Fig. 6a). Several genes were differentially expressed, including IL-22, and genes that are downstream of it, such as *Reg3* genes and digestive enzymes also produced by the jejunum (Fig. 6b). The KEGG pathway analysis of WT and *KSR23* jejunum indicated that genes involved in protein digestion and absorption, such as chymotrypsinogen B1 (*Ctrb1*), trypsinogen (*Prss1*), trypsin (*Tyr5* and *Tyr4*), caboxypeptidase a1 (*Cpa1*), and chymotrypsin-like elastase family member 3b (*Cela3b*), were downregulated in the intestine of *KSR23* mice (Fig. 6c). Transcriptome analysis also showed decreased expression of several genes involved in the regulation of the very-low-density lipoprotein particle pathway (Fig. 6d, e). These particles regulate fat and cholesterol release into the bloodstream. Together, these results indicate that systemic expression of IL-23, and other cytokines, correlated with significant transcriptional changes in genes that regulate food processing in the pancreas and intestine.

**IL-22 ablation corrects aberrant pancreatic enzyme gene expression**. To ask if IL-22 affected the expression of pancreatic enzymes, we examined the expression of several genes in the pancreas of *KSR23* and *KSR23/IL-22⁻/⁻* mice. qPCR confirmed that *KSR23* mice had increased messenger RNA (mRNA) levels of *Reg3β* (Fig. 7a) and decreased levels of pancreatic lipase, amylase, and *Sycn* in the pancreas than that in WT controls (Fig. 7c, g). Changes in protein levels were also documented by immunohistochemistry (Fig. 7b, h). Deletion of IL-22 in *KSR23* mice had a marked effect on the levels of expression of these markers, as well as on the levels of pSTAT3, a known mediator of IL-22

signaling[18] (Fig. 7i). Levels of *Reg3β*, amylase, and pancreatic lipase, as well as *Sycn* in the pancreas of *KSR23/IL-22⁻/⁻*, mice were equivalent to those observed in WT mice, suggesting that IL-22, or genes downstream of it, had a major role in controlling expression of pancreatic genes.

**IL-22 directly regulates expression of pancreatic genes**. Deletion of IL-22 reverts the two main phenotypes of early death and stunted body growth in animals expressing IL-23. In addition, deletion of IL-22 reverted the abnormalities in gene expression observed in the pancreas. IL-22R is expressed by acinar cells in the pancreas of WT mice, and it has been shown that IL-22 can directly promote expression of *Reg3β* by these cells[19]. To investigate if IL-22 could directly affect expression of pancreatic genes involved in nutrient processing, we performed in vitro experiments. Short-term (24 h) incubation of WT pancreatic cells with IL-22 led to a significant upregulation of *Reg3b* as reported[19], but no changes in expression of *Ptf1a*, *Sycn*, *Amy2a*, and *Amy2b* (Supplementary Fig. 6). Longer incubation of the acinar cells with IL-22 (60 h) led to a significant decrease in the expression of *Ptf1a* (Fig. 8). Coincident with the decreased expression of *Ptf1a*, we observed a significant decrease in the expression of *Sycn*, *Amy2a*, and *Prss2* in cultures stimulated with IL-22 (Fig. 8), suggesting a direct role for IL-22 in the regulation of these pancreatic genes. To confirm that these changes were elicited by IL-22 signaling, we studied the effect of IL-22 in pancreatic cells derived from IL-10R2-deficient mice. No changes in gene expression were observed after the addition of IL-22 to IL-10R2 signaling-deficient pancreatic acinar cells (Fig. 8 and Supplementary Fig. 6), confirming that the changes observed in the expression of pancreatic genes were mediated by the IL-22 receptor. Together, these results

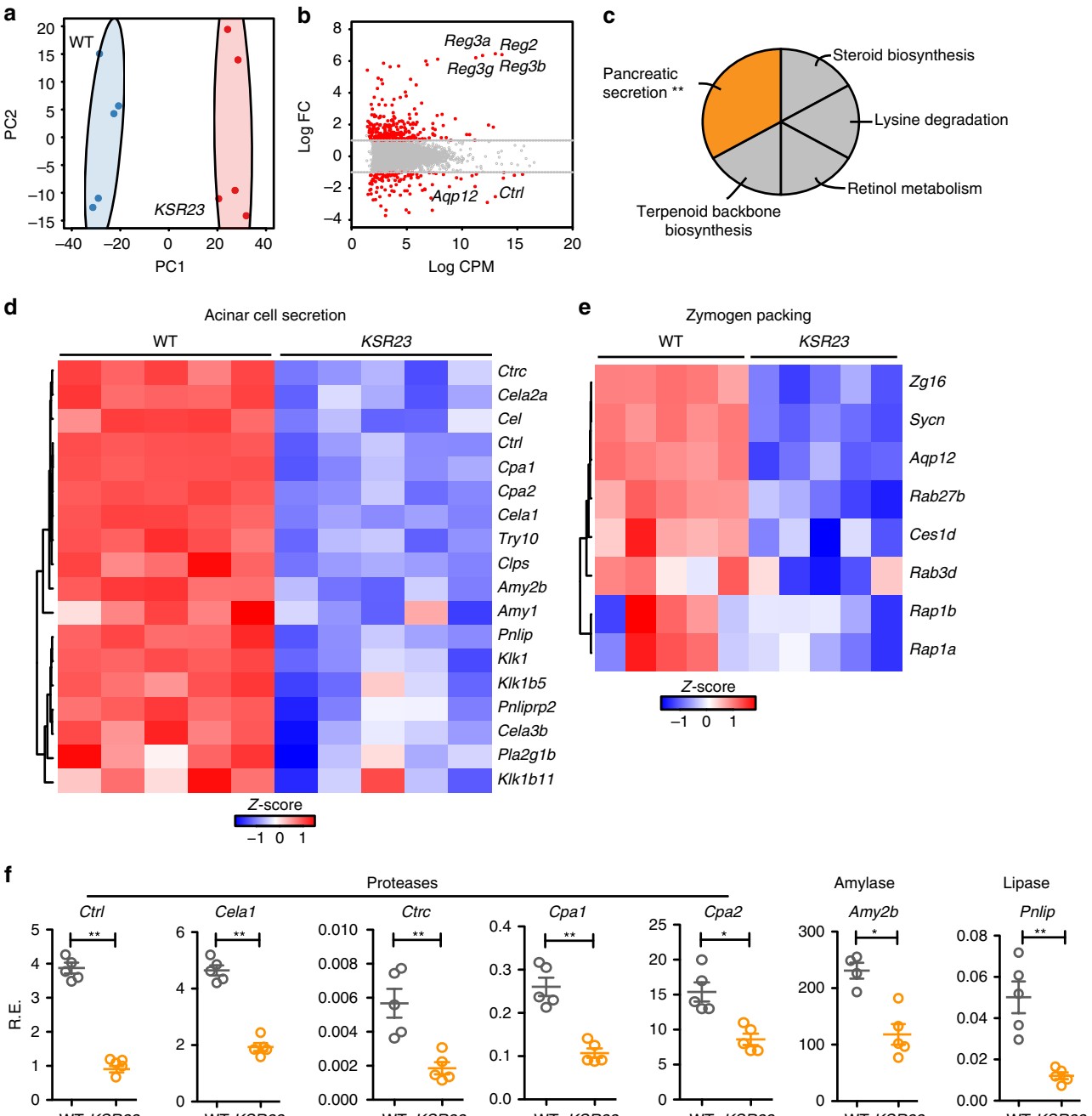

**Fig. 4** Decreased pancreatic secretion pathway in the pancreas of *KSR23* mice. **a** Principal component analysis of RNASeq expression data from all biological replicates of WT and *KSR23* pancreas (*n* = 5/group). **b** Plot of log 2 FC (log 2 fold change) vs. log 2 CPM (log 2 counts per million) of all detected transcripts. Points are colored according to expression status: non-significant genes (gray) and significant genes (385 genes; *Q* < 0.05; log 2 FC > 1 or log 2 FC < −1; red). **c** The KEGG pathway analysis of significantly downregulated genes in the pancreas of *KSR23* mice (*Q* < 0.05; terms >3 genes; % Genes/term >4; κ 0.4). **d** *Z*-scored heat map of downregulated genes associated with the GO terms "pancreatic secretion" and acinar secretory enzymes. **e** The heat map of downregulated genes associated with the term zymogen packing. **f** qPCR validation of genes encoding digestive enzymes. Data are shown as mean ± SEM, statistical analysis by nonparametric Mann–Whitney test. *\*p* < 0.05 and *\*\*p* < 0.01. Source data are provided as a Source Data file

indicate that the inhibition of pancreatic gene expression observed in *KSR23* mice is mediated by IL-22.

**Decreased expression of pancreatic enzymes in NEC mice.** The phenotype of neonatal death presented by the *V23*[11] and *CXR23* mice is reminiscent of that presented by humans with necrotizing enterocolitis (NEC), a disease characterized by bowel necrosis and multisystem organ failure in the neonatal period[20]. NEC mostly

affects premature infants and involves the invasion of the intestinal wall by bacteria, which could lead to dysregulation of IL-23 production. To test this hypothesis, we induced NEC in C57Bl/6 mice at postnatal day 7, following protocols described by Hackam and colleagues[21,22]. qPCR analysis showed that expression of IL-23 and IL-22 was upregulated in the terminal ileum of animals with NEC when compared to controls at P11 (Fig. 9a, b). As systemic level of IL-22 were upregulated in the *CXR23* and *KS23* mice, we tested next if the increased levels of IL-23 observed

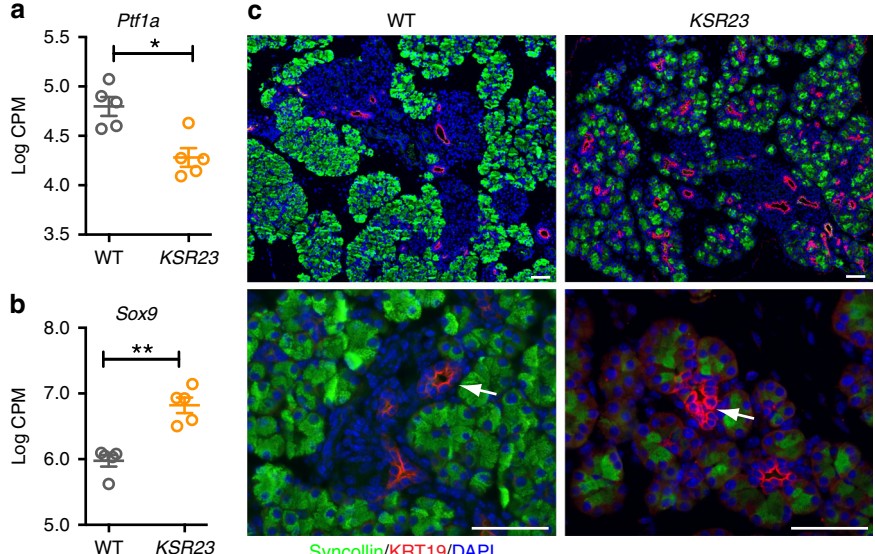

**Fig. 5** Increased number of ductal cells in the pancreas of *KSR23* mice. **a**, **b** RNASeq analysis of the expression of *Ptf1a* (**a**) and *Sox9* (**b**) in the pancreas of WT and *KSR23* mice at P5 ($n = 5$ mice/group). Data are shown as mean ± SEM, and statistical analysis by nonparametric Mann–Whitney test. *$p < 0.05$ and **$p < 0.01$. **c** Immunostaining of pancreas of WT and *KSR23* mice with antibodies against syncollin and KRT19. Arrows show localization of KRT19-positive cells in the pancreas of WT and *KSR23* mice. Scale bars = 50 µm. Source data are provided as a Source Data file

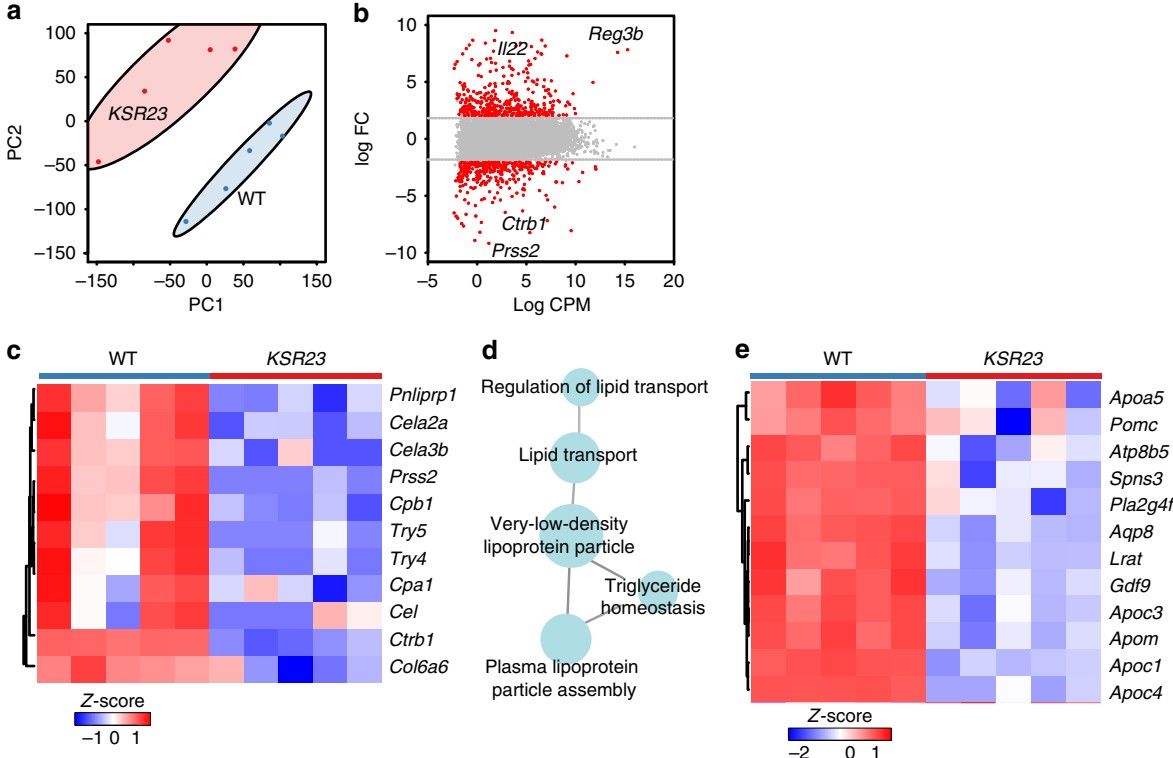

**Fig. 6** Altered food absorption pathways in the jejunum of *KSR23* mice. **a** Principal component analysis of RNASeq expression data from all biological replicates of WT and *KSR23* jejunum ($n = 5$/group). **b** Plot of log 2 FC (log 2 fold change) vs. log 2 CPM (log 2 counts per million) of all detected transcripts. Points are colored according to expression status: non-significant genes (gray) and significant genes (698 genes; $Q < 0.05$; log 2 FC > 2 or log 2 FC < −2; red). **c** Z-scored heat map of all downregulated genes associated with the GO terms "protein digestion and absorption." **d** ClueGo analysis of significantly downregulated genes shown enrichment of GO term group "very-low-density lipoprotein particle." **e** Z-scored heat map of all downregulated genes associated with the GO group "very-low-density lipoprotein particle"

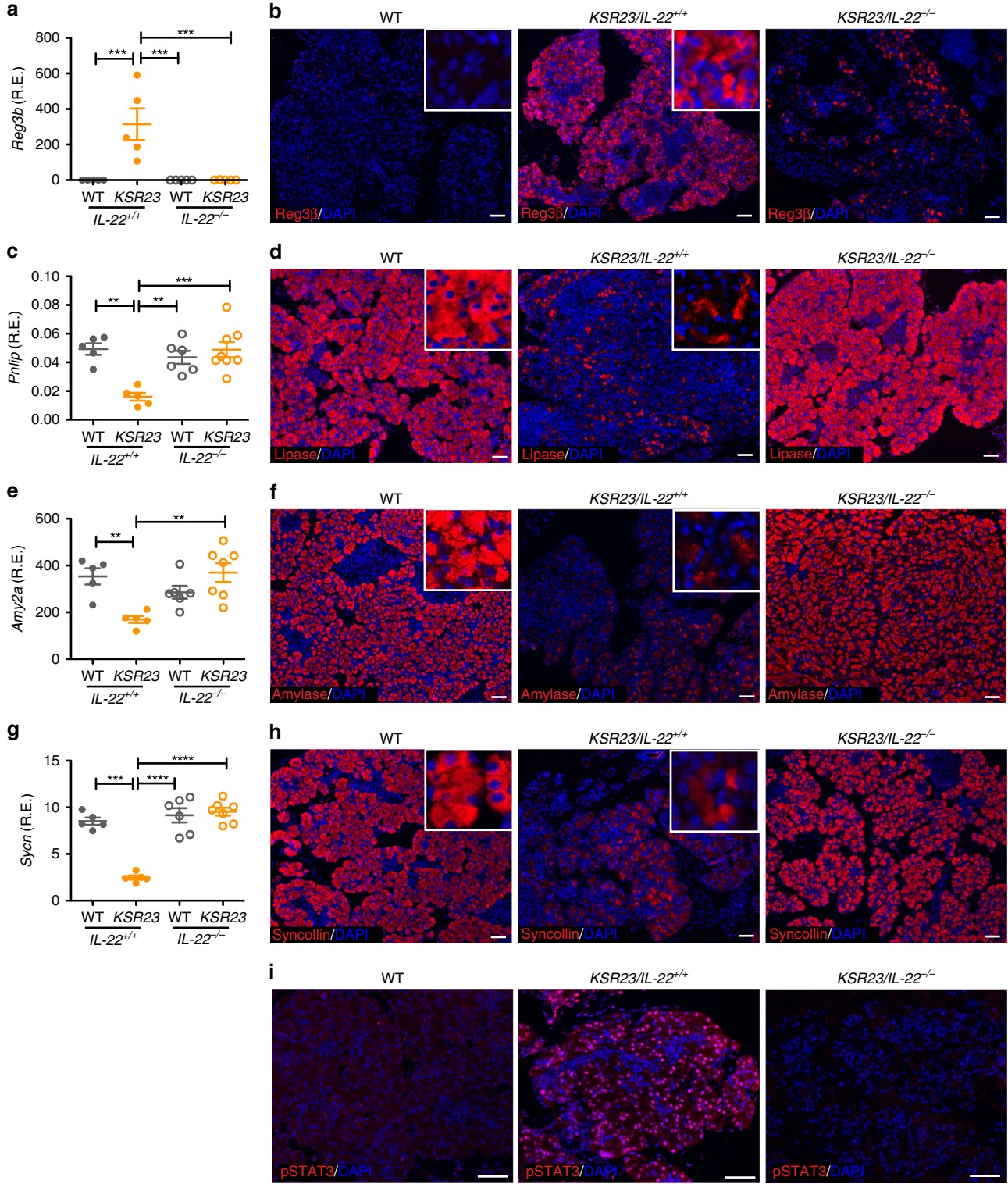

**Fig. 7** Ablation of IL-22 restores expression of pancreatic genes in *KSR23* mice. **a**, **c**, **e**, **g** qPCR analysis of the expression of selected genes in the pancreas of WT, *IL-22⁻/⁻*, *KSR23, and KSR23/IL-22⁻/⁻* mice at P5. **b**, **d**, **f**, **h** Immunostaining of pancreas of WT, *KSR23*, and *KSR23/IL-22⁻/⁻* mice at P5 with antibodies against *Reg3β* (**b**), lipase (**d**), amylase (**f**), and syncollin (**h**). Insets show higher magnification. **i** Immunostaining of pSTAT3 in the pancreas of WT, *KSR23*, and *KSR23/IL-22⁻/⁻* mice. Increased number of pSTAT3⁺ cells in the pancreas of *KSR23* mice. Data are shown as mean ± SEM, statistical analysis by one-way ANOVA. **\*\****p* < 0.01 and **\*\*\****p* < 0.001. Scale bars = 50 μm. Source data are provided as a Source Data file

in the intestine of mice with NEC were associated with increased levels of IL-22 in circulation. As shown in Fig. 9c, we noted a significant increase in circulating levels of IL-22 in mice with NEC compared to controls. We also tested if the expression of

pancreatic enzymes, *Sycn* and *Reg3β*, was altered in the pancreas of mice with NEC. Using immunostaining we detected a marked increase in the expression of *Reg3β* and decreased expression of amylase A, lipase, *Sycn* in the pancreas of animals with NEC

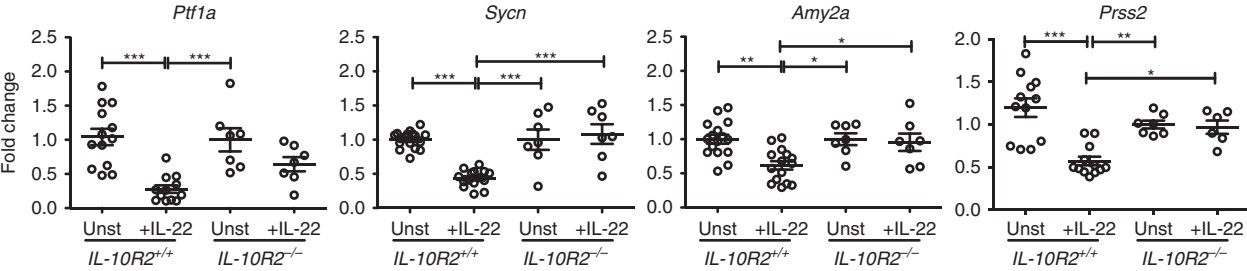

**Fig. 8** IL-22 induces downregulation of pancreatic enzymes related genes in acinar cell cultures in vitro. Acinar cells from WT (*IL-10R2+/+*) and *IL-10R2−/−* mice were cultured in the presence of IL-22 recombinant protein. Sixty hours after culture, cells were analyzed for the expression of pancreas transcription factor 1a (*Ptf1a*), syncollin (*Sycn*), amylase 2a (*Amy2a*), and serine protease 2 (*Prss2*) by qPCR. Note the downmodulation of several pancreatic related genes by WT acinar cells cultured with IL-22. IL-22 did not reduce the expression of pancreatic related genes in IL-10R2 -deficient cells. Data are shown as mean ± SEM, statistical analysis by one-way ANOVA. $*p < 0.05$, $**p < 0.01$, and $***p < 0.001$. Source data are provided as a Source Data file

(Fig. 9d, e). Together, the results indicate that expression of IL-23 and IL-22 are increased in the intestine in the NEC model of neonatal inflammation, and that these changes are associated with dysregulation in the expression of pancreatic enzymes, mimicking what is observed in our transgenic models.

## Discussion

Here we study the impact of IL-23 expression in newborns. We report that IL-23 induces severe phenotypes when expressed since birth by CX3CR1+ cells and by keratinocytes. These phenotypes are reversed by inactivation of IL-22, suggesting that IL-22 mediates the pathogenic properties of IL-23 in newborns. We also show that a critical and unappreciated pathogenic mechanism triggered by IL-22 is the reduction in the expression of several pancreatic enzymes, which contributes to the failure to thrive observed in these animals.

Two phenotypes were associated with neonatal expression of IL-23. The first was death within the first 48 h of life. This phenotype was present in ~50% of the mice expressing IL-23 in CX3CR1+ myeloid cells in the intestine, but not in mice expressing IL-23 in keratinocytes, and was associated with intestinal bleeding, resembling what is observed in animals expressing IL-23 from the villin promoter (*V23* mice)[11]. The phenotype was less severe in the case of *CXR23* mice, perhaps reflecting lower levels of IL-23 expression compared with *V23* mice. The cause of death in both cases is not clear, but the microbiota could play an important role, as GF *CXR23* mice do not present intestinal bleeding and survive beyond the neonatal period. Also similar to what is observed in the *V23* mice[11], we observed an increase in the number of ILC3 in the intestine of *CXR23* mice. We suggest that this cell population is the key driver of pathogenesis in *CXR23* mice, similar to what has been demonstrated in the *V23* mice. Given that ILC3 produce copious amounts of IL-22 upon IL-23 stimulation, it is likely that they are a major source of IL-22 in the newborn gut. Increased IL-22 levels could disrupt intestinal permeability, favor permeation of bacteria into the lamina propria, and cause marked intestinal inflammation and intestinal bleeding and death[11]. This hypothesis is supported by our observation that inactivation of IL-22 signaling (using both IL-22 and IL-10R2-knockout (KO) mice) rescues the intestinal bleeding observed in 50% of the *CXR23* mice. In this context it is important to discuss that IL-10R2 mediates signaling of both IL-22 and IL-10. The fact that the KO of IL-10R2 in the context of IL-23 expression phenocopies that elicited by KO of IL-22 further strengthens the hypothesis of a pathogenic role for IL-22, and suggests that IL-10 signaling does not affect the phenotype of premature death induced by Il-23 expression. Together, our data support a pathogenic role of IL-22 in the neonatal mice with increased IL-23 expression.

Another finding that merits discussion is the fact that the newly acquired microbiota appears to contribute to the development of the intestinal bleeding phenotype observed in *CXR23* SPF neonates. We show here that *CXR23* GF mice do not develop intestinal bleeding. This finding suggests that either bacteria have a direct role in the process or that they control the overall number of CX3CR1+ cells in the intestine (and therefore regulate the levels of IL-23 being produced). Indeed, it was shown that the number of CX3CR1+ cells is reduced in the lamina propria of GF adult mice[23]. To test if the difference in phenotype between *CXR23* SPF and *CXR23* GF mice is due to a reduced number of CX3CR1+ cells in the SI of the GF mice, we performed flow cytometry. We found that at birth (P1) *CXR23* SPF and *CXR23* GF mice have similar number of CX3CR1+ cells in the SI (SPF $7 ± 0.6 × 10^6$ vs. GF $6 ± 0.6 × 10^6$). We conclude that the differences in phenotype cannot be ascribed to the number of CX3CR1+ cells in the SI and that the microbiota contributes directly to the demise of the mice at this stage. We hypothesize that they may do so by infiltrating the intestinal wall. We have shown previously that intestinal permeability is increased in areas immediately adjacent to the ILC3-rich lymphoid anlagen in the SI of mice expressing IL-23 in intestinal epithelial cells[11].

The second phenotype observed in both strains was failure to growth. *CXR23* mice that survived the first 48 h failed to grow and died prematurely, similar to *KSR23* mice. Again, these phenotypes were reversed by IL-22 ablation, indicating a pathogenic role for elevated levels of IL-22. Our results are in agreement with other findings in the literature. Transgenic expression of IL-22 from the EµLCK promoter or from the insulin promoter results in animals with small body weight that die a few days after birth[24]. Expression of IL-22 from the albumin promoter, on the other hand, does not induce early lethality, but the body weight of the animals is lower than that of WT littermates from 5 months of age[25]. The reason why animals in this line survive, and the other strains driven by the EµLCK and insulin[24], villin[11], and CX3CR1 promoters do not is unclear, but it might be related to overall levels of IL-22 in circulation or the site and timing of its expression. Low body weight is also observed in mice in which IL-22 was delivered by adenoviral infection or in animals injected daily with this cytokine (25 mg over a 2-week period). Daily injection of IL-22 induced a 8% decrease in body weight[26]. The causes of stunted growth and low body weight are multiple and include reduced absorption of dietary components. Indeed, it was suggested previously that IL-22 regulates expression of lipid transporters in the intestine[27], a finding that we confirm and extend here.

The most unexpected finding in the course of our experiments was that increased levels of IL-22 in circulation markedly affected

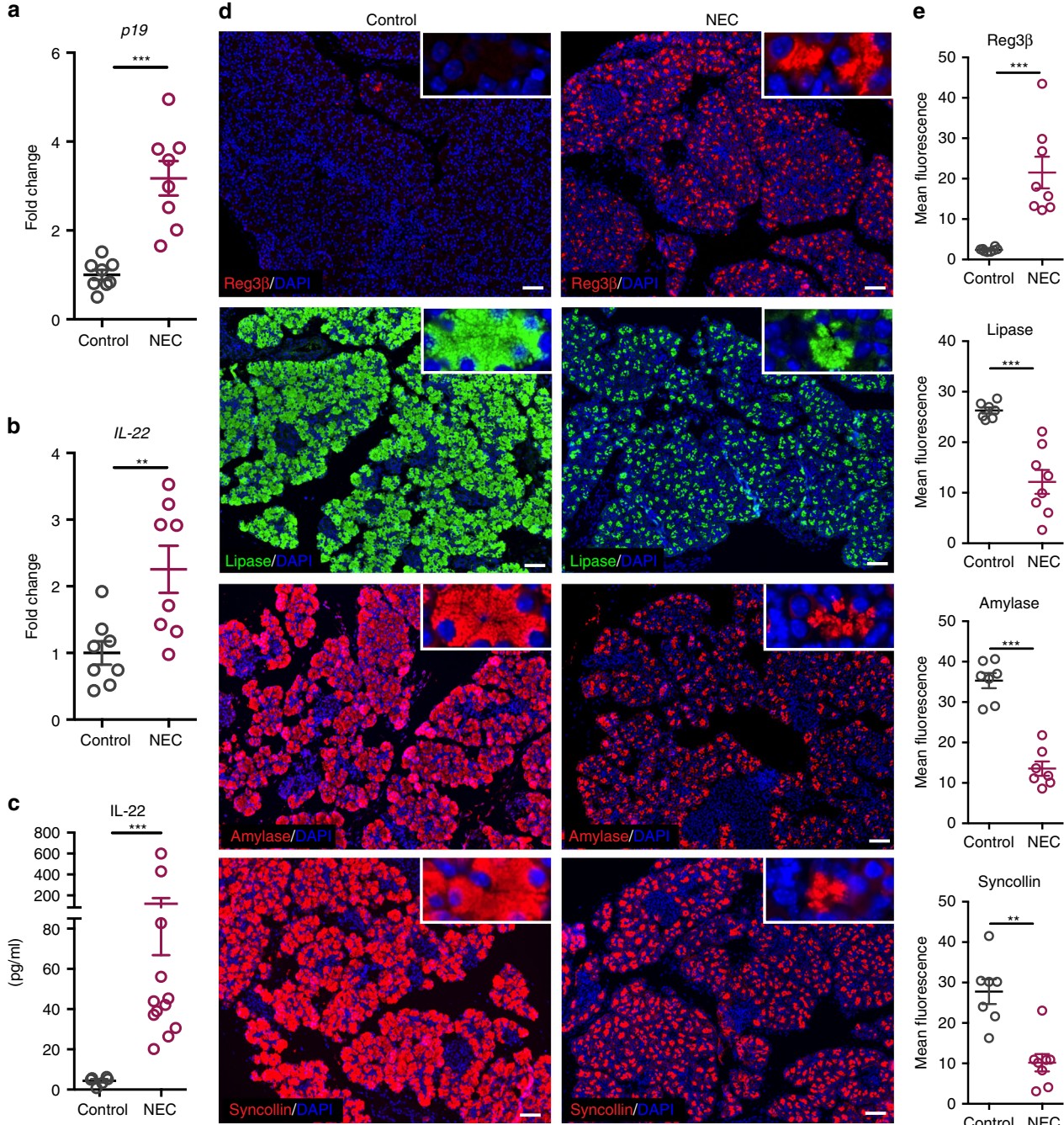

**Fig. 9** Mice with necrotizing enterocolitis (NEC) show decreased production of pancreatic enzymes. **a–c** Expression of IL-23p19 (**a**) and IL-22 (**b**) mRNA in the terminal ileum of control and animals with NEC ($n = 8$ mice/group). **c** Quantification of IL-22 in the serum of control and animals with NEC ($n = 6$–12 mice/group). **d** Representative immunostaining of pancreas of control mice and mice with NEC. Insets show higher magnification ($n = 7$–8 mice/group). Scale bars = 50 μm. **e** Quantification of fluorescence intensity in images shown in **d** ($n = 7$–8 mice/group). Data are shown as mean ± SEM, statistical analysis by nonparametric Mann–Whitney test. **$p < 0.01$ and ***$p < 0.001$. Source data are provided as a Source Data file

the expression of pancreatic enzymes critical for digestion of carbohydrates, proteins, and lipids. The reduced expression of enzymes and reduced absorption of nutrients by the intestine resulted in malabsorption, as evidenced by increased amount of fat in the stools. The expression of the digestive enzymes was normalized by deletion of IL-22. Thus, it is likely that the growth disturbances observed in the *CXR23* and *KSR23* mice result in part from malabsorption of critical dietary components due to reduced expression of pancreatic enzymes and of intestinal transporters.

Most tissue-specific genes known to be regulated by IL-22 are upregulated. Of particular interest, we have identified genes that are downregulated by IL-22. While we cannot rule out that the other cytokines simultaneously upregulated by IL-23 may impact the expression of the pancreatic and intestinal genes, our demonstration that IL-22 can directly downregulate expression of genes encoding the pancreatic enzymes in vitro suggests that IL-22 is the main factor regulating this process in IL-23-expressing animals. We show here that incubation of acinar cells with IL-22 leads to a significant reduction in the expression of *Ptf1a* after a

long (60 h) but not short (20 h) incubation. Ptf1a is a basic helix–loop–helix transcription factor that is critical for the specialized phenotype of the pancreatic acinar cells, including the control of the production of secretory digestive enzymes[17]. We show an increase in the number of pSTAT3 acinar cells of the *KSR23* mice and postulate that this may be the effector mechanism for the reduced expression of digestive enzymes. Interestingly, the genetic ablation of Ptf1a in adults leads to reduction in the expression of genes encoding acinar secretory products and genes involved in the zymogen granule formation[16]. Also of interest is the observation that decreased Ptf1a expression results in increased expression of ductal cell markers[16,17]. Similar results were observed in *KSR23* mice. Our observations linking high systemic levels of IL-22 with reduced levels of Ptf1a expression in the pancreas of *KS23* mice, and the demonstration of a direct activity of IL-22 on the acinar cell, strongly support a role for IL-22 in the expression of *Ptf1a*. Given the kinetics of the response in vitro and the increased number of ductal cells in vivo in transgenic mice, we cannot rule out that the reduction in the expression of pancreatic genes is due to a developmental defect induced by lower expression of Ptf1a.

The IL-22-induced upregulation of several antimicrobial genes by acinar cells is yet another interesting observation in this context. IL-22 directly regulates expression of *Reg3β* in vitro, as shown here and elsewhere[19]. The additional secretion of the microbicide Reg proteins into the intestinal lumen could potentially affect the intestinal microbiota. Thus, the fact that IL-22 can regulate expression of several pancreatic genes may have implications for the pathogenesis of other diseases of the intestinal tract. Varying degrees of pancreatic insufficiency have been reported in patients with inflammatory bowel disease (IBD)[28–30], and these may be functionally related to higher levels of IL-22 in situ or in circulation. Relevant to this discussion is the observation that pediatric IBD is often associated with growth failure[31], which is caused by several factors, among them decreased food intake and malabsorption[32].

NEC is a severe inflammatory disease that affects the intestine of premature babies leading to intestinal ischemia and death[20]. Of interest, expression of IL-22 is increased in the intestine of humans with NEC[22]. To test whether the IL-23/IL-22 axis was altered in NEC, we performed RNA quantitation in the affected intestinal tissue (terminal ileum) and observed significantly elevated levels of both cytokines. Levels of IL-22 in circulation were elevated in the mice with NEC and there was upregulation of *Reg3β* and downregulation of pancreatic enzymes and *Sycn*. These results closely mimic the results obtained with transgenic animals and support the hypothesis that increased expression of IL-23 in lesional tissue results in increased local and systemic levels of IL-22. We hypothesize that the increased systemic levels of IL-22 in both the transgenic and experimental NEC settings are critical to affect pancreatic expression of *Reg3β*, *Sycn*, and the pancreatic enzymes.

In summary, we have demonstrated an important role for IL-23 in neonatal pathology. The main effector mechanism triggered by IL-23 is increased production of IL-22, which acts at the level of the intestinal epithelium and acinar cells in the pancreas to regulate intestinal permeability and inhibit mechanisms associated the digestion and absorption of nutrients. These results indicate that dysregulation in the expression of IL-23 and IL-22 has a significant negative impact on survival and overall fitness of newborns.

## Methods

**Mice**. C57BL/6 (CN 000664), CX3CR1-cre[13] (CN 025524), hK14-cre[33] (CN 018964), and IL-10R2-KO[34] (CN 005027) mice were purchased from The Jackson laboratory (Bar Harbor, ME). IL-22tdTomato mice[23] were provided by Dr. Scott

Durham (NIH, Bethesda, MD). *R23* mice were described previously[12]. *R23* mice were backcrossed into the C57BL/6 background for 11 generations. Mice were maintained under specific pathogen-free conditions. The GF mice were bred in-house and were housed in standard flexible film isolators in our GF animal facility. All animal experiments in this study were approved by the Institutional Animal Care and Use Committee of Icahn School of Medicine at Mount Sinai, and were performed in accordance with the approved guidelines for animal experimentation at the Icahn School of Medicine at Mount Sinai.

**Generation of IL-22-deficient mice**. $IL-22^{-/-}$ mice were generated using CRISPR/Cas9 technology directly into C57BL/6 as described before[35]. To do so, we designed a single guide RNA (sgRNA) that targeted IL-22 exon 1 by using the online CRISPR Design Tool (http://tools.genome-engineering.org). The plasmid expressing sgRNA was prepared by ligating oligos into *Bbs*I site of pX330 (Addgene plasmid ID: 42230)[36]. Confirmation of WT and KO genotypes was done by sequencing. Alignment of amino acid sequence between truncated protein generated in IL-22-KO mice and IL-22 from WT mice[37] was done by SeqBuilder. Functional validation of the IL-22-KO was done as described in Supplementary Fig. 5.

**Induction of NEC**. NEC was induced as described[21,22,38,39] in 7-day-old mouse pups by gavage feeding (five times per day for 4 days) of formula [Similac Advance infant formula (Abbott Nutrition) and Esbilac canine milk replacer (PetAg) at a ratio of 2:1] supplemented with enteric bacteria that were isolated from an infant with NEC. Mice were exposed to brief hypoxia (5% $O_2$, 95% $N_2$ for 10 min twice daily) for 4 days. All animals were euthanized on day 11. All experiments and procedures were approved by the Johns Hopkins University Animal Care and Use committees in accordance to the Guide for the Care and Use of Laboratory Animals (8th Edition, The National Academies Press 2011).

**Immunostaining, imaging, and quantification**. Organs were dissected, fixed in 10% phosphate-buffered formalin, and processed for paraffin sections. Four-micrometer sections were dewaxed and hydrated. Antigen retrieval was performed by microwaving tissue sections for 15 min in Target Retrieval Solution (DAKO). Tissue sections were incubated overnight with primary antibodies (Abs) in a humidified atmosphere at 4 °C. After washing, conjugated secondary Abs were added and then incubated for 35 min. The slides were washed and mounted with Fluoromount-G (SouthernBiotech). Primary and secondary Abs used are listed in Supplementary Table 1. Immunofluorescence imaging was performed in a fluorescence microscope (Nikon Eclipse Ni) with Plan Apo objective lenses. Images were acquired using a digital camera (DS-QiMc; Nikon) and Nis-Elements BR imaging software. After setting the acquisition conditions according the brightest sample, all the acquisition parameters were kept the same at all times, including the laser intensity, the exposure time, the gain of the photomultiplier, the offset of the histogram, and the image magnification. An average of three determinations were made for each mouse examined. The fluorescence intensity was measured by Image J/Fiji. Images were composed in Adobe Photoshop CS3.

**Flow cytometry**. The SI of P1 mice were microdissected, minced, and digested with 2 mg ml$^{-1}$ collagenase D (Roche). Cell suspensions were passed through a 70-μm cell strainer and mononuclear cells were isolated. Cells were pre-incubated with anti-mouse CD16/CD32 for blockade of Fc receptors, and incubated with the appropriate monoclonal Ab conjugates. Propidium iodide (Sigma-Aldrich) or DAPI (4′,6-diamidino-2-phenylindole) (Invitrogen) was used to distinguish live cells from dead cells during cell analysis. Stained cells were analyzed on a FACS Canto or LSRII machine using the Diva software (BD Bioscience). Data were analyzed with FlowJo software (TreeStar). ILC3 cells were defined as CD45$^+$Lin$^-$Thy1$^+$Sca-1$^{hi}$ as described[11]. The following fluorochrome-conjugated anti-mouse antibodies were used at indicated dilutions: Thy-1.2 (53-2.1, 1:200), Sca-1 (D7, 1:200), CD45 (30-F11, 1:200), CD11b (M1/70, 1:200), IL-22 (1H8PWSR, 1:100), IL-17A (eBio17B7, 1:100) were from eBioscience; lineage cocktail (CD3, B220, CD11b, Gr-1, Ter119$^-$) (145-2C11, RA3-6B2, M1/70, RB6-8C5, TER-119, 1:100) and CX3CR1 (SA011F11, 1:200) were from BioLegend.

**Enzyme-linked immunoabsorbent assay**. Chemokines and cytokines (granulocyte-colony-stimulating factor,, GM-CSF, chemokine (C-X-C motif) ligand 1 (CXCL1), CXCL2, CXCL10, IFNγ, IL-1α, IL-1β, IL-12p70, IL-15/IL-15R, IL-17, IL-18, IL-21, IL-22, IL-23, IL-25, IL-27, leptin, C-C motif chemokine ligand 2 (CCL2), CCL3, CCL4, CCL5, CCL7, macrophage colony-stimulating factor, soluble receptor activator of nuclear factor-κB ligand, tumor necrosis factor) were measured in mouse serum with ProcartaPlex Multiplex Immunoassays (eBioscience) according to the manufacturer's protocol. Analysis was performed with the xMAP Technology by Luminex. The levels of IL-22 in the serum of mice with NEC were measured with a IL-22 Mouse ELISA Kit (Thermo Fisher Scientific) according to the manufacturer's protocol.

**Triglycerides**. Blood was collected from WT and *KSR23* mice at P5 after a 4 h starvation period. Serum separation was performed using the Capillary Blood

Collection Tubes (Terumo™ Capiject™). Serum TG levels were measured with the Triglyceride (GPO) reagent set (Pointe Scientific, Inc.) according to the manufacturer's specifications. To measure TGs in the stool, we collected 1 mg of feces in a 1.5 ml Eppendorf tube and resuspended it in 500 μl of phosphate-buffered saline (PBS). Five hundred microliters of chloroform in methanol solution (2:1) was added to the fecal suspension and vortexed. The suspension was centrifuged (1000 × $g$, 10 min, at room temperature) and the lower liquid phase, containing the extracted lipids in chloroform:methanol, was transferred to a new 1.5 ml Eppendorf tube. Samples were incubated for 24 h at 37 °C in order to evaporate all liquid. The lipid pellet was then resuspended in 200 μl PBS and the TG concentration measured with the Triglyceride (GPO) reagent set (Pointe Scientific, Inc.) according to the manufacturer's specification.

**Acinar cell culture**. Pancreatic acinar cells were isolated as described[40] and cultured for 24 and 60 h in Waymouth's medium alone or with addition of recombinant IL-22 (50 ng/ml, Kingfisher Biotech) at 37 °C, 5% $CO_2$.

**Reverse-transcription polymerase chain reaction**. Total RNA from tissues was extracted using the RNeasy Mini/Micro Kit (Qiagen) according to the manufacturer's instructions. Complementary DNA (cDNA) was generated with Superscript III (Invitrogen). qPCR was performed using SYBR Green Dye (Roche) on the 7500 Real-Time System (Applied Biosystems) machine. Results were normalized to the housekeeping gene *Ubiquitin*. Relative expression levels were calculated as $2^{(Ct (Ubiquitin) - Ct(gene))}$. Primers were designed using Primer3Plus and are listed in Supplementary Table 2. For detection the expression of the genes listed in Supplementary Table 2 in NEC mice, qPCR were performed by using the Bio-Rad CFX96 Real-Time System (Bio-Rad) and was measured relative to the housekeeping gene *RPLO*.

**RNA sequencing**. SI and pancreas were homogenized in Trizol reagent (Invitrogen). Total RNA from tissues was extracted using the RNeasy Mini Kit (Qiagen) according to the manufacturer's instructions. Samples were shipped on dry ice to the Center for Functional Genomics and the Microarray and HT Sequencing Core Facility at the University at Albany (Rensselaer). RNA quality was assessed using the NanoDrop (Thermo Scientific) and Bioanalyzer Total RNA Pico assay (Agilent). Total RNA with a RNA integrity number value of 8 or greater was deemed of good quality to perform the subsequent protocols. One hundred picograms of total RNA was oligo-dT primed using the SMART-Seq v4 Ultra Low Input RNA Kit (Clontech) and the resulting cDNA was amplified using 15 cycles of PCR. The double-stranded cDNA (dscDNA) was purified using AMPure XP magnetic beads and assessed for quality using the Qubit dsDNA HS assay and an Agilent Bioanalyzer high-sensitivity dscDNA chip (expected size ~600–9000bp). The Illumina Nextera XT Kit was used for library preparation wherein 125 pg dscDNA was fragmented and adaptor sequences added to the ends of fragments following which 12 cycles of PCR amplification was performed. The DNA library was purified using AMPure XP magnetic beads and final library assessed using Qubit dsDNA HS assay for concentration and an Agilent Bioanalyzer high-sensitivity DNA assay for size (expected range ~600–740 bp). Library quantitation was also done using a NEBNext Library Quant Kit for Illumina. Each library was then diluted to 4 nM, pooled, and denatured as per standard Illumina protocols to generate a denatured 20 pM pool. A single-end 75 bp sequencing was performed on the Illumina Nextseq 500 by loading 1.8 pM library with 5% PhiX on to a 75 cycle high output flow cell. The RNASeq data was checked for quality using the Illumina FastQC algorithm on Basespace.

**Transcriptome analyses**. RNASeq data from small intestine and pancreas was mapped to the mouse reference genome (UCSC/mm10) using Tophat version 2.1.0[41]. Gene-level sequence counts were extracted for all annotated protein-coding genes using htseq-count version 0.6.1[42] by taking the strict intersection between reads and the transcript models associated with each gene. Raw count data were filtered to remove low expressed genes with less than five counts in any sample. Differentially expressed genes between groups were analyzed using Bioconductor EdgeR package version 3.10.2 Bioconductor/R[43,44]. Statistically significant differentially expressed genes between groups ($Q < 0.05$) were selected in gene-wise log-likelihood ratio tests that were corrected for multiple testing by Benjamini and Hochberg false discovery rate. KEGG pathway enrichment analyses were performed using ClueGo[45,46] to identify pathways in which differentially expressed genes are involved. A cut-off of 0.4 was set for κ-score and terms including at least three genes were retrieved.

**Statistical analyses**. Except for RNASeq data, differences between groups were analyzed with nonparametric Mann–Whitney test or one-way analysis of variance (ANOVA). For repeated-measures analysis, ANOVA with Bonferroni post hoc test was used. Survival curves were analyzed by a log-rank test. All statistical analyses were performed with GraphPad Prism 7 software (GraphPad, La Jolla, CA). Differences were considered significant when $p < 0.05$. Data are shown as mean ± SEM unless specified otherwise.

**Reporting summary**. Further information on research design is available in the Nature Research Reporting Summary linked to this article.

## Data availablity

All RNASeq data generated are available at Gene Expression Omnibus (GEO) under the accession number GSE137415. The authors declare that the data that support the findings of this study are available within the paper and its Supplementary Information files. Extra data or information are available from the corresponding authors upon request. The raw data underlying Figs. 1, 2, 3, 4, 5, 7, 8, 9 and Supplementary Figs. 3–6 are available in the Source Data file.

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

## Acknowledgements

We thank Dr. Maria C. Lafaille for comments, Dr. Ana Valbuena for help with the Luminex assay, and Dr. Kevin Kelley and the Mouse Genetics and Gene Targeting CoRE Facility for assistance in generation of gene-modified mice. This work was supported by a grant from the National Institutes of Health (R01DK110352) to S.A.L. L.C. was supported by a Career Development Award (634253) from the Crohn's and Colitis Foundation of America (CCFA).

## Author contributions

L.C., V.S., Z.H., M.D., J.C.-D., T.M.M., T.K., C.P.S., D.J.H. and G.C.F. did experiments and analyzed data. J.J.F., S.K.D. and H.X. provided reagents. G.C.F. and S.A.L. designed study, analyzed data, and wrote the manuscript. All authors reviewed and edited the manuscript.

## Competing interests

The authors declare no competing interests.

## Additional information

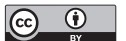 

