## [Peer Review File · Nature Communications]

Reviewers' comments:

Reviewer #1, immunometabolism expert (Remarks to the Author):

The manuscript by Glaucia C. Furtado and colleagues investigates the pathologic mechanisms underlying neonatal inflammatory diseases, which are known to be associated with severe morbidity. The general approach used is to enforce the expression of the pro-inflammatory cytokine IL-23 constitutively in CX3CR1+ myeloid cells or in and keratinocytes. In both cases the authors find that these animals fail to grow and die prematurely. The authors also found that premature death is associated with dysregulation of the expression of molecules involved in food digestion and absorption, in pancreas and intestine, leading to a malabsorptive condition that is typical of neonatal inflammatory diseases. Mechanistically, the pathogenic effect of IL-23 was functionally linked to the induction of IL-22, which signals via the IL-22R in pancreatic acinar cells to inhibit the expression of the pancreas associated transcription factor 1a (Ptf1a). The authors conclude that neonatal induction of the IL-23/IL-23 axis contributes to the malabsorption state that promotes the pathological outcome of neonatal inflammatory diseases. The manuscript is technically sound and well presented and the comments listed above are more conceptual than technical.

Major comments

Comment #1: The overexpression approach used by the authors provides a proof-of-principle that IL-23 can act via IL-22 to disrupt key physiologic functions that likely contribute to the pathogenesis of neonatal inflammatory diseases and their associated morbidity. Specifically, the authors show that IL-23 overexpression by CX3CR1+ myeloid cells or by keratinocytes is sufficient per se to trigger some of the hallmarks of this disease. However, they fail to provide a clear rationale why they chose to overexpress IL-23 in either one of these cell compartments and how this relates to the pathogenesis of neonatal inflammatory diseases. For example, is there expression of IL-23 by CX3CR1+ myeloid cells or keratinocytes under pathophysiologic conditions where neonatal inflammatory diseases develop? If so, can one prevent the lethal outcome of this disease in mice using a loss-of-function approach in which CX3CR1+ myeloid cells or keratinocytes fail to express IL-23. A similar reasoning can be made for the contribution of the IL-22/IL-22R to the pathogenesis of this disease. This type of argumentation in the introduction and eventual addition of experimental work, using a loss-of-function approach, would provide a clear pathophysiologic context to the findings reported.

Comment #2: The observation that genetic ablation of IL-22 or IL-10R2 reduces lethality and restores normal body growth in neonates expressing IL-23 (Figure 3) is impressive. This is followed on Figures 4-6 by an equally impressive set of findings showing that IL-22/IL22R impairs food absorption. If this is indeed the underlying cause of the pathology that develops, these mice maybe rescued by exogenous nutrient administration. It is not clear that this experiment can be achieved technically but if so this would be very interesting to test

Reviewer #2, IL-22 expert (Remarks to the Author):

In this manuscript by Furtado et al., the authors study the effects of ectopically expressed inflammatory cytokine in neonatal mice. This is an extension of a previous study where the authors generated mice in which IL-23 was expressed in the GI tract under the villin promoter. After birth, these mice quickly died due to severe intestinal pathology. In this new study, the authors generated mice expressing IL-23 under the control of the CX3CR1 or Keratin14 promoter, driving cytokine expression in myeloid cells and keratinocytes, respectively. They find that these mice are again sickly and die soon after birth. They further show that IL-22, a cytokine known to be downstream of IL-23

signaling, is required for this phenomenon. IL-22 is well known to have modulatory effects on mucosal and other tissues. Their most interesting finding is that IL-23 downregulates several pancreatic enzymes, suggesting that pathology in the IL-23 overexpression mice is driven by increased levels of IL-22 downregulating important digestive enzymes, leaving the neonatal mice with poor regulation of food processing and nutrient acquisition. In vitro experiments complement the in vivo data and show that stimulation of acinar cells with IL-22 down regulates several key genes.

Studies such as these where scientists generate genetically modified mice that overexpress a cytokine confound me. The premise is that you can study cytokine biology by studying aberrant expression, even if it doesn't model any known human disease. This approach is somewhat reasonable upon initial discovery of a cytokine, but IL-23 biology is well described. This makes the phenotype of these mice somewhat unsurprising, especially as the group has previously generated similar mice. Nevertheless, there are some valuable findings in this manuscript. This study is of particular interest as it has identified genes that are downregulated by the cytokine IL-22. Most tissue-specific genes known to be regulated by IL-22, are upregulated by the cytokines. This could be better stressed in the discussion. Secondly, less is known on the role of IL-22 in the pancreas. IL-22 is dual-natured and has been both shown to be protective or inflammatory in different inflammatory models. However, known protective roles vastly outnumber inflammatory, so this study is a welcome addition to the IL-22 literature. Lastly, the experiments are well controlled and properly analyzed, which contribute to the high readability of this manuscript.

Specific points

1. There is clearly a role for microflora in their mice as they found that when they generated germ free mice the could extend their lifespan from days to weeks. As their findings suggest that IL-23-mediated upregulation of IL-22 leads to downregulation of key pancreatic enzymes, can life span also be extended if mice are provided a broken-down minimal diet?
2. Are low homeostatic levels of IL-22 required for normal pancreatic function? The authors do not examine IL-22 deficient mice. Past studies have showed that some IL-22 regulated genes are highly regulated by IL-22 (for example, RegIIIg) and are found at very low levels in IL-22 KO mice. Have the authors described what not only an inflammatory role for IL-22, but also a role in health?
3. Supplemental Figure 4 describing generation of an IL-22 deficient mouse via CRISPR/Cas9 technology would benefit from the codons being marked on the figure. At what point in the hypothetical protein is a stop codon introduced?
4. The authors may want to consider adding Supplemental Figure 5 to their main manuscript.

Reviewer #3, gastrointestinal expert (Remarks to the Author):

The study by Furtado et al provides some potentially very important novel findings in relation to the pathological role of excess IL-23-driven IL-22 in the intestine and pancreas of neonates. The experiments appear carefully conducted, with appropriate controls and statistical approaches. Much of the data are derived from artificial models of overexpression of IL-23 in either myeloid cells or keratinocytes, and it could be argued that these models don't mimic well real biology. However, certain exposures in early life may drive high local IL-23 in the gut and these models could well reflect on the mechanisms at play. One such condition is necrotising enterocolitis in newborns/premature neonates (see specific comment). I have provided below some suggested areas to provide further data/description of the models, and it would be particularly beneficial to have a deeper understanding of the pancreatic effect to understand whether it is a direct effect on differentiated cells or a change in differentiation pathways.

Major Points

1. In CCR23 mice the localisation of IL-23 expressing CX3CR+ cells in the small intestine and colon should be described including whether the expression of IL-23 changes the density of these cells. Similarly, IL-23 expression and pathology in the small intestine should be described in more detail

with respect to localisation from duodenum to jejunum, and comment should be made as to whether any pathology is seen in the colon.

2. In KSR23 mice the level of IL-23 in the GIT, including the oral mucosa and esophagus should also be assessed. In Fig 2I were serum triglycerides measured after a period of fasting or were mice fed?

3. In the germ free (GF) CXR23 mice it would be good to explore the levels of both IL-23 and the downstream cytokines in blood and tissues vs normally colonised mice. It may be that there are fewer CX3CR+ cells in the intestine under GF conditions.

4. The experiment reported in Fig 3 where IL10R2 is knocked out will also block IL-10 signalling. Whilst one would predict that would have a pro-inflammatory effect, it should be acknowledged that this is the case. Crossing with a IL-22R1 ko would give a more specific deletion, but given there is also an experiment with an IL-22 ko giving the same result, acknowledging this caveat is sufficient.

5. The data on gene expression after exposure of acinar cell cultures to IL-22 in Fig 7 are based on a 60 h post-IL-22 exposure, whereas after 12 h they report no change (and this is not shown). This result requires explanation given IL-22 drives STAT3-driven transcriptional changes within minutes/early hours of exposure. Together with the altered differentiation states reported in the pancreas in vivo suggests the effects could be on progenitor cells or differentiating cells in these cultures rather than changing expression of digestive enzyme genes in well differentiated cells.

Minor Points

1. Figure 1 and legend – K – this is reported to be the number of ILC3+ cells in the small intestine – is this the entire small intestine or a segment, and has the number been corrected for weight/length of small intestine utilised? The legend should read ILC3 ($\times 10^{-3}$) if the mean is ~ 6000 cells/intestine ie. $6 = 6000 \times 10^{-3}$

2. There was no discussion of the possible role of IL-22 in pathology in necrotising enterocolitis which is an important complication in premature human infants in which IL-23 and ILC's have been implicated. It may be worth making this possible connection.

Reviewer #1, immunometabolism expert (Remarks to the Author):

The manuscript by Glaucia C. Furtado and colleagues investigates the pathologic mechanisms underlying neonatal inflammatory diseases, which are known to be associated with severe morbidity. The general approach used is to enforce the expression of the pro-inflammatory cytokine IL-23 constitutively in CX3CR1+ myeloid cells or in and keratinocytes. In both cases the authors find that these animals fail to grow and die prematurely. The authors also found that premature death is associated with dysregulation of the expression of molecules involved in food digestion and absorption, in pancreas and intestine, leading to a malabsorptive condition that is typical of neonatal inflammatory diseases. Mechanistically, the pathogenic effect of IL-23 was functionally linked to the induction of IL-22, which signals via the IL-22R in pancreatic acinar cells to inhibit the expression of the pancreas associated transcription factor 1a (Ptf1a). The authors conclude that neonatal induction of the IL-23/IL-23 axis contributes to the malabsorption state that promotes the pathological outcome of neonatal inflammatory diseases. The manuscript is technically sound and well presented and the comments listed above are more conceptual than technical.

Major points

1. The overexpression approach used by the authors provides a proof-of-principle that IL-23 can act via IL-22 to disrupt key physiologic functions that likely contribute to the pathogenesis of neonatal inflammatory diseases and their associated morbidity. Specifically, the authors show that IL-23 overexpression by CX3CR1+ myeloid cells or by keratinocytes is sufficient per se to trigger some of the hallmarks of this disease. However, they fail to provide a clear rationale why they chose to overexpress IL-23 in either one of this cell compartments and how this relates to the pathogenesis of neonatal inflammatory diseases. For example is there expression of IL-23 by CX3CR1+ myeloid cells or keratinocytes under pathophysiologic conditions where neonatal inflammatory diseases develop? If so, can one prevent the lethal outcome of this disease in mice using a loss-of-function approach in which CX3CR1+ myeloid cells or keratinocytes fail to express IL-23. A similar reasoning can be made for the contribution of the IL-22/IL-22R to the pathogenesis of this disease. This type of argumentation in the introduction and eventual addition of experimental work, using a loss-of-function approach, would provide a clear pathophysiologic context to the findings reported.

Answer: We thank the reviewer for the comments and suggestions. Please find below responses to the general comments

1.1 However, they fail to provide a clear rationale why they chose to overexpress IL-23 in either one of this cell compartments

Answer: The rationale for expressing IL-23 in CX3CR1+ cells was that these are the cells in which IL-23 is normally expressed, and are enriched in the intestine. We hypothesized that expression of IL-23 in this compartment would be lower than that we obtained when we expressed it using the villin promoter (all epithelial cells in the gut), and therefore, could result in viable offspring, allowing further insights into IL-23 biology in the intestine. The rationale to express it in the skin was to test if the localization of IL-23 to a different body segment could elicit similar phenotypes (in this case premature death and stunted growth). The rationale is described in the new text, please see below:

"The immaturity of the immune system in the neonatal phase is thought to account for the increased infectious morbidity and mortality shown by newborns as compared to older children and adults. During this phase, the immaturity of the adaptive response is partially compensated by heightened responses of innate immune cells to bacterial stimuli. One such example is the increased response of neonatal dendritic cells to signals conveyed by toll-like receptor (TLR) ligation¹. Neonatal dendritic cells exposed to TLR4 ligands produce high levels of IL-23^{2, 3, 4}, an important regulator of myeloid and innate lymphoid cell biology. Deficit in IL-23 signaling is not associated with a major deficit in the immune function in newborn mice, but its expression has been detected in the intestine of rat models of necrotizing enterocolitis, a severe inflammatory condition affecting neonates^{5, 6, 7}. It is not clear if under these conditions IL-23 has protective or pathogenic functions, but overexpression of IL-23 in the neonatal gut of transgenic mice promotes severe intestinal inflammation, intestinal bleeding and perinatal death⁸.

To further understand the biological role of IL-23 expression in early life, and its contribution to intestinal inflammation we generated two novel mouse strains in which IL-23 was overexpressed in CX3CR1-positive myeloid cells⁹ and keratinocytes. We chose to express IL-23 in myeloid cells, the cells that normally produce it in the intestine, to gain further insight into its possible role in intestinal inflammatory conditions. To assess if IL-23 expressed in a remote site could also affect intestinal function, we chose to express it in keratinocytes."

1.2 and how this relates to the pathogenesis of neonatal inflammatory diseases.

Answer: Please see above.

1.3 For example is there expression of IL-23 by CX3CR1+ myeloid cells or keratinocytes under pathophysiologic conditions where neonatal inflammatory diseases develop?

Answer: Unfortunately there is no data in the literature regarding the cell origin of IL-23 in experimental models of NEC in rats. However, given the fact that bacteria seem to be critical for development of NEC, given that DC and other myeloid cells (that express CX3CR1) are involved in the response, and given that DCs have a heightened ability to express IL-23 during the neonatal phase, it is likely that intestinal DCs and macrophages express IL-23 during models of intestinal inflammation in neonates. The expression of IL-23 in the skin is not associated with NEC and was used here to test if remote production of IL-23 could lead to an inflammatory phenotype in the gut.

1.4 For example is there expression of IL-23 by CX3CR1+ myeloid cells or keratinocytes under pathophysiologic conditions where neonatal inflammatory diseases develop? If so, can one prevent the lethal outcome of this disease in mice using a loss-of-function approach in which CX3CR1+ myeloid cells or keratinocytes fail to express IL-23.

Answer: We have searched for experimental models in which IL-23 or IL-22 are increased in mouse neonates but have not find a suitable one. Most of the models of NEC are in rats and the ones in mice cannot be performed adequately given the difficulty of gavaging newborn mice. Thus, we could not directly test the relevance of IL-23 or IL-22 using loss-of-function models.

2. The observation that genetic ablation of IL-22 or IL-10R2 reduces lethality and restores normal body growth in neonates expressing IL-23 (Figure 3) is impressive. This is followed on Figures 4-6 by a equally impressive set of findings showing that IL-22/IL22R impairs food absorption. If this is indeed the underlying cause of the pathology that develops, these mice maybe rescued by exogenous nutrient administration. It is not clear that this experiment can be achieved technically but if so this would be very interesting to test.

Answer: We agree with the reviewer that providing exogenous nutrient administration would help determine if food impairment is indeed the underlying cause of the pathology. We have made multiple attempted to gavage mice before they are 10 days old, but have not been successful.

Reviewer #2, IL-22 expert (Remarks to the Author):

In this manuscript by Furtado et al., the authors study the effects of ectopically expressed inflammatory cytokine in neonatal mice. This is an extension of a previous study where the authors generated mice in which IL-23 was expressed in the GI tract under the villin promoter. After birth, these mice quickly died due to severe intestinal pathology. In this new study, the authors generated mice expressing IL-23 under the control of the CX3CR1 or Keratin14 promoter, driving cytokine expression in myeloid cells and keratinocytes, respectively. They find that these mice are again sickly and die soon after birth. They further show that IL-22, a cytokine known to be downstream of IL-23 signaling, is required for this phenomenon. IL-22 is well known to have modulatory effects on mucosal and other tissues. Their most interesting finding is that IL-22 downregulates several pancreatic enzymes, suggesting that pathology in the IL-23 overexpression mice is driven by increased levels of IL-22 downregulating important digestive enzymes, leaving the neonatal mice with poor regulation of food processing and nutrient acquisition. In vitro experiments complement the in vivo data and show that stimulation of acinar cells with IL-22 down regulates several key genes.

Studies such as these were scientists generate genetically modified mice that overexpress a cytokine confound me. The premise is that you can study cytokine biology by studying aberrant expression, even if it doesn't model any known human disease. This approach is somewhat reasonable upon initial discovery of a cytokine, but IL-23 biology is well described. This makes the phenotype of these mice somewhat unsurprising, especially as the group has previously generated similar mice. Nevertheless, there are some valuable findings in this manuscript. This study is of particular interest as it has identified genes that are downregulated by the cytokine IL-22. Most tissue-specific genes known to be regulated by IL-22, are upregulated by the cytokines. This could be better stressed in the discussion. Secondly, less is known on the role of IL-22 in the pancreas. IL-22 is dual-natured and has been both shown to be protective or inflammatory in different inflammatory models. However, known protective roles vastly outnumber inflammatory, so this study is a welcome addition to the IL-22 literature. Lastly, the experiments are well controlled and properly analyzed, which contribute to the high readability of this manuscript.

Answer: We thank the Reviewer for the positive assessment of our work and for pointing out that the manuscript is a “welcome addition to the IL-22 literature”. The

Reviewer also points out that: *“This study is of particular interest as it has identified genes that are downregulated by the cytokine IL-22. Most tissue-specific genes known to be regulated by IL-22, are upregulated by the cytokines. This could be better stressed in the discussion”*. We have introduced a commentary to this in our revised discussion. Kindly see page 17, lines 382-383 in our revised manuscript.

Major Points

1. *There is clearly a role for microflora in their mice as they found that when they generated germ free mice that could extend their lifespan from days to weeks. As their findings suggest that IL-23-mediated upregulation of IL-22 leads to downregulation of key pancreatic enzymes, can life span also be extended if mice are provided a broken-down minimal diet?*

Answer: We agree that this would be a good approach, but as discussed above (also see answer 2 for the reviewer 1), our attempts to gavage mice before they were 10 days old have not been successful.

2. *Are low homeostatic levels of IL-22 required for normal pancreatic function? The authors do not examine IL-22 deficient mice. Past studies have showed that some IL-22 regulated genes are highly regulated by IL-22 (for example, RegIIIg) and are found at very low levels in IL-22 KO mice. Have the authors described what not only an inflammatory role for IL-22, but also a role in health?*

Answer: The reviewer raises an interesting question regarding the role of IL-22 normal pancreatic function. Our studies indicate that ablation of IL-22 in neonatal mice does not affect expression of genes encoding pancreatic enzymes. We show in **Figure 7 (Panels a, c, e and g** in our revised manuscript) that there are no differences in the expression of pancreatic genes (e.g. Reg3b, Pnlip, Amy2a, Sycn) between WT and IL-22 knock-out mice during the neonatal stage. Accordingly, IL-22 KO mice grew normally compared to WT mice. Whether IL-22 has a role in normal pancreatic function in the adult, and whether increased levels of IL-22 observed in inflammatory conditions affect pancreatic function in the adult remain open questions.

3. *Supplemental Figure 4 describing generation of an IL-22 deficient mouse via CRISPR/Cas9 technology would benefit from the codons being marked on the figure. At what point in the hypothetical protein is a stop codon introduced?*

Answer: We thank the reviewer for this suggestion. The mutation introduced in the IL-22 gene generated a premature STOP signal, which theoretically should have produced a truncated peptide if translated. This peptide lacks the ability to bind to the IL-22 receptor, according to published information¹⁰. We included this information in the new **Supplemental Figure 4**.

4. *The authors may want to consider adding Supplemental Figure 5 to their main manuscript.*

Answer: We thank the reviewer for this suggestion. We have changed it from Supplemental Figure 5 to **Figure 7i** in the revised manuscript.

Reviewer #3, gastrointestinal expert (Remarks to the Author):

The study by Furtado et al provides some potentially very important novel findings in relation to the pathological role of excess IL-23-driven IL-22 in the intestine and pancreas of neonates. The experiments appear carefully conducted, with appropriate controls and statistical approaches. Much of the data are derived from artificial models of overexpression of IL-23 in either myeloid cells or keratinocytes, and it could be argued that these models don't mimic well real biology. However, certain exposures in early life may drive high local IL-23 in the gut and these models could well reflect on the mechanisms at play. One such condition is necrotizing enterocolitis in newborns/premature neonates (see specific comment). I have provided below some suggested areas to provide further data/description of the models, and it would be particularly beneficial to have a deeper understanding of the pancreatic effect to understand whether it is a direct effect on differentiated cells or a change in differentiation pathways.

Answer: We thank the reviewer for the positive assessment of our work and for pointing out that: “*certain exposures in early life may drive high local IL-23 in the gut and these models could well reflect on the mechanisms at play. One such condition is necrotizing enterocolitis in newborns/premature neonates*”. We mention this interesting link in birth the Introduction (pages 4 & 5, lines 78-112) and Discussion (page 18, lines 415-420) of our revised manuscript.

Major Points

1. In CXR23 mice the localization of IL-23 expressing CX3CR1+ cells in the small intestine and colon should be described including whether the expression of IL-23 changes the density of these cells. Similarly, IL-23 expression and pathology in the small intestine should be described in more detail with respect to localization from duodenum to jejunum, and comment should be made as to whether any pathology is seen in the colon.

Answer: We found that the number of CX3CR1+ cells in the small intestine of wild type mice was higher than that found in the large intestine at birth. At this point the number of CX3CR1+ cells in the small intestine of CXR23 mice was 2 fold higher than that found in the small intestine of controls. These results are shown in **new Figure 1 (Panels j and k)**. The erosive lesions in the small intestine of CXR23 mice were similar to those observed in animals expressing IL-23 from the villin promoter (V23 mice)⁸. The lesions were localized in the Peyer's patches (PP) Anlagen and present only in the small intestine (duodenum, jejunum). In addition, we observed occasional extravasation of erythrocytes from villi. No pathology was observed in the colon or other tissues. Kindly see page 6 and 7 lines 134-146 in our revised manuscript and **Supplemental Figure 1**.

2. In KSR23 mice the level of IL-23 in the GIT, including the oral mucosa and esophagus should also be assessed. In Fig 2I were serum triglycerides measured after a period of fasting or were mice fed?

Answer: Levels of IL-23 in the oral mucosa and esophagus were assessed as requested and are now part of **Figure 2 (Panel c)**. In regards to serum triglyceride

measurements, we describe in the **Methods**, under **Triglycerides** (page 20, lines 473-474 in our revised manuscript), that blood was collected from WT and *KSR23* mice at P5 after a 4hr starvation period.

3. In the germ free (GF) *CXR23* mice it would be good to explore the levels of both IL-23 and the downstream cytokines in blood and tissues vs normally colonized mice. It may be that there are fewer CX3CR+ cells in the intestine under GF conditions.

Answer: The reviewer raises the hypothesis that the differences in outcomes in *CXR23* mice raised in germ-free (GF) condition may be due to a reduced number of CX3CR1+ cells, as there is evidence that the number of these cells are lower in adult GF mice¹¹. To directly address this question, we measured the number of CX3CR1 cells in the small intestine of newborn GF mice and compared it to that found in specific-pathogen-free (SPF) mice at postnatal day 1. As shown in **Fig 1 for Reviewers**, there is no difference in the number of CD45+ and CX3CR1+ cells in the immediate neonatal period. Therefore, the differences in phenotype cannot be ascribed to reduced number of CX3CR1+ cells in the GF mice at this time point. We have introduced additional discussion on this subject in our revised manuscript (page 15, lines 339-354 in our revised manuscript).

Figure 1 for Reviewers: Number of CD45+ and CX3CR1+ cells in the small intestine of SPF and GF mice at postnatal day 1. Data are shown as mean ± sem, n = 4-5 mice/group.

4. The experiment reported in Fig 3 where *IL10R2* is knocked out will also block IL-10 signalling. Whilst one would predict that would have a pro-inflammatory effect, it should be acknowledged that this is the case. Crossing with a *IL-22R1 ko* would give a more specific deletion, but given there is also an experiment with an *IL-22 ko* giving the same result, acknowledging this caveat is sufficient.

Answer: We thank the Reviewer for the suggestion. The fact that the deletion of the *IL10R2* would in theory increase proinflammatory activity is mentioned in the new text. "Crossing with a *IL-22R1 ko* would give a more specific deletion, but given there is also an experiment with an *IL-22 ko* giving the same result, acknowledging this caveat is sufficient". This is now acknowledged (pages 14 & 15, lines 329-337 in our revised manuscript).

5. The data on gene expression after exposure of acinar cell cultures to IL-22 in Fig 7 are based on a 60 h post-IL-22 exposure, whereas after 12 h they report no change (and this is not shown). This result requires explanation given IL-22 drives STAT3-driven transcriptional changes within minutes/early hours of exposure. Together with the altered differentiation states reported in the pancreas in vivo suggests the effects could be on progenitor cells or differentiating cells in these cultures rather than changing expression of digestive enzyme genes in well differentiated cells.

Answer: The reviewer raises an important point. We have now adjusted the discussion to indicate that given the kinetics of the response and the increased number of ductal cells in vivo in transgenic mice, we cannot rule out that the reduction in the expression of pancreatic genes is due to a developmental defect (page 17, lines 387-389 and 400-402 in our revised manuscript).

Minor Points

1. Figure 1 and legend – K – this is reported to be the number of ILC3+ cells in the small intestine – is this the entire small intestine or a segment, and has the number been corrected for weight/length of small intestine utilized? The legend should read ILC3 ($\times 10^{-3}$) if the mean is ~6000 cells/intestine ie. $6 = 6000 \times 10^{-3}$.

Answer: We thank the reviewer for raising this issue, and apologize for the misunderstanding. The number of ILC3 shown in the **Figure 1** should be the number of cells per whole small intestine of mouse at P1. We have included additional details in the figure legend.

2. There was no discussion of the possible role of IL-22 in pathology in necrotizing enterocolitis which is an important complication in premature human infants in which IL-23 and ILC's have been implicated. It may be worth making this possible connection.

Answer: We thank the Reviewer for helping us to improve our manuscript. We have included this consideration in the discussion. Kindly see page 18, lines 415-420 in our revised manuscript.

Reviewer #4 (Remarks to the Author)

In the manuscript by Furtado and colleagues, the authors generated and characterized IL-23 transgenic mice either specifically expressed in myeloid cells or in keratinocytes. In both cases, these animals died prematurely although at different time. Significant pathological and transcriptome changes have been observed in the intestine and pancreas of these animals. IL-22 is a key downstream cytokine driven the pathology since genetic ablation of IL-22 increase the life span of these animals. IL-22 can directly acts on pancreatic acinar cells to affect the gene expression and function. This work is a follow up study of previous IL-23 transgenic mice published by the same group in which the IL-23 was specifically expressed in the intestine. Several issues need to be further addressed.

1) *The novelty of this study is compromised a little by the previous study, although the authors did provide some additional mechanistic insights. One of the major issues here is the authors used three different transgenic lines (including one in the previous paper) with IL-23 expressed in different tissues. However, phenotypically it was not clear which*

phenotypes if any distinguished these lines due to the tissue specificity of IL-23 expression. The level of systemic IL-23 is different from these lines. Is this sufficient to explain all the different phenotypes observed. Either yes or not the authors should provide sufficient data and discussion.

Answer: The main phenotypes described in our manuscript are premature death and stunted growth. Animals from both strains described in this manuscript die before 15 days of age. In the case of the animals expressing IL-23 from the CX3CR1+ cells we observed ~50% mortality by 3 days of age. The perinatal death associated with intestinal bleeding is similar to that observed in mice expressing IL-23 from intestinal epithelial cells (V23 mice)⁸. We suspect that the phenotype of intestinal bleeding is function of the levels of IL-23 in the intestine, and that it is more striking in the case of the V23 mice, because they express higher levels of IL-23 locally. The levels of IL-23 systemically are lower in mice expressing IL-23 in the skin, so we cannot rule out that phenotype of perinatal death is associated with the levels of IL-23 systemically. The other phenotype is stunted growth. In all three transgenic strains deletion of IL-22 rescues the phenotypes of premature death and stunted growth, clearly indicating a pathogenic role for this cytokine. Data on the deletion of IL-22 in V23 mice was not added to the manuscript originally and is added here for the reviewer (**Figure 2 for Reviewers**).

Figure 2 for Reviewers: Survival curves of V23 and V23/IL-22^{-/-} mice. $P < 0.0001$, by a log-rank test.

2) Following the above point, the authors should better organize the manuscript, especially two transgenic lines. Currently, it is not well integrated and rationale for why making two is just not clear.

Answer: We have made the changes as requested. The rationale is described in the new text (pages 4 & 5, lines 78-112 in our revised manuscript).

3) Does IL-22 fully rescue all the phenotypes of the transgenic mice? Any phenotypes left?

Answer: CXR23 mice that survive beyond day 3, and KSR23 mice, share a phenotype of stunted growth a premature death (before 3 weeks of age). These two phenotypes were rescued by ablation of IL-22. The study of additional phenotypes is underway.

4) From Figure 4-7 the authors specifically followed the KSR23 mice. Are these phenotypes only observed in KSR23 but not the other two mouse line? If yes, what is the underline mechanism? Or simply those mice die too early to examine? It is not unclear by reading the manuscript.

Answer: The animals in both transgenic strains perish before the third week of life.

Because *KSR23* animals survive marginally longer we used them for our mechanistic studies. Please see **Figure 1d** and **Figure 2f** in our revised manuscript.

5) *Any differences observed between IL-22^{-/-} and IL-10R2^{-/-} mice, since IL-10R2 also abolished other cytokine pathways such as IL-10.*

Answer: Our analysis focused on the role of IL-22 and its receptor on phenotypes of stunted growth and premature death elicited by IL-23. No additional information on these animals is available at present. Please see pages 14 & 15, lines 329-337 in our revised manuscript.

Citations

1. Corbett NP, Blimkie D, Ho KC, Cai B, Sutherland DP, Kallos A, *et al.* Ontogeny of Toll-like receptor mediated cytokine responses of human blood mononuclear cells. *PLoS one* 2010, **5**(11): e15041.
2. Yerkovich ST, Wikstrom ME, Suriyaarachchi D, Prescott SL, Upham JW, Holt PG. Postnatal development of monocyte cytokine responses to bacterial lipopolysaccharide. *Pediatric research* 2007, **62**(5): 547-552.
3. Kollmann TR, Crabtree J, Rein-Weston A, Blimkie D, Thommai F, Wang XY, *et al.* Neonatal innate TLR-mediated responses are distinct from those of adults. *Journal of immunology* 2009, **183**(11): 7150-7160.
4. Vanden Eijnden S, Goriely S, De Wit D, Goldman M, Willems F. Preferential production of the IL-12(p40)/IL-23(p19) heterodimer by dendritic cells from human newborns. *European journal of immunology* 2006, **36**(1): 21-26.
5. Dvorak K, Coursodon-Boydiddle CF, Snarrenberg CL, Kananurak A, Underwood MA, Dvorak B. Helicobacter hepaticus increases intestinal injury in a rat model of necrotizing enterocolitis. *American journal of physiology Gastrointestinal and liver physiology* 2013, **305**(8): G585-592.
6. Underwood MA, Arriola J, Gerber CW, Kaveti A, Kalanetra KM, Kananurak A, *et al.* Bifidobacterium longum subsp. infantis in experimental necrotizing enterocolitis: alterations in inflammation, innate immune response, and the microbiota. *Pediatric research* 2014, **76**(4): 326-333.

7. Coursodon-Boydiddle CF, Snarrenberg CL, Adkins-Rieck CK, Bassaganya-Riera J, Hontecillas R, Lawrence P, *et al.* Pomegranate seed oil reduces intestinal damage in a rat model of necrotizing enterocolitis. *American journal of physiology Gastrointestinal and liver physiology* 2012, **303**(6): G744-751.
8. Chen L, He Z, Slinger E, Bongers G, Lapenda TLS, Pacer ME, *et al.* IL-23 activates innate lymphoid cells to promote neonatal intestinal pathology. *Mucosal immunology* 2015, **8**(2): 390-402.
9. Chen L, He Z, Iuga AC, Martins Filho SN, Faith JJ, Clemente JC, *et al.* Diet Modifies Colonic Microbiota and CD4(+) T Cell Repertoire to Induce Flares of Colitis in Mice With Myeloid-cell Expression of Interleukin 23. *Gastroenterology* 2018.
10. de Moura PR, Watanabe L, Bleicher L, Colau D, Dumoutier L, Lemaire MM, *et al.* Crystal structure of a soluble decoy receptor IL-22BP bound to interleukin-22. *FEBS letters* 2009, **583**(7): 1072-1077.
11. Niess JH, Adler G. Enteric flora expands gut lamina propria CX3CR1+ dendritic cells supporting inflammatory immune responses under normal and inflammatory conditions. *Journal of immunology* 2010, **184**(4): 2026-2037.

Reviewers' comments:

Reviewer #2 (Remarks to the Author):

The authors have addressed my concerns. I only raise these very minor points.

Page 7, line 142

As the authors do not actually show IL-22 and/or IL-17 production from the ILC3s, they should add "potentially" or another modifier before "capable of producing..."

Figure 1 j,k and n, and Page 15 lines 348-9

Another reviewer brought up this issue and it has not been resolved. I believe the y-axis is incorrect and should be 10^4 not 10^{-4} (j and k) and 3 for n. On page 15 there is also confusion as to total cell number.

Reviewer #3 (Remarks to the Author):

The authors have made reasonable efforts to address the issues that I and the other reviewers made on the original version of the manuscript, and the manuscript is substantially improved.

Areas which I believe remain somewhat unclear include:

1. How critical the deficiency in pancreatic enzyme production is to the phenotype (wasting and death). The authors claim it is not feasible to address this as gavage is not feasible in neonatal mice.
2. Whether the alterations in the exocrine pancreas reflect differentiation changes in cells evolving from stem cells rather than de-differentiation of acinar cells. The data, including some new data, seems to favour the former and may be an effect of IL-22 only applicable to neonates.

Having highlighted these issues, the manuscript has some caveats around these two points now so that is more acceptable.

Prof Michael McGuckin

Reviewer #4 (Remarks to the Author):

The authors largely addressed by comments in the revised manuscript.

Reviewer #2 (Remarks to the Author):

The authors have addressed my concerns. I only raise these very minor points.

Comment 1: Page 7, line 142. As the authors do not actually show IL-22 and/or IL-17 production from the ILC3s, they should add "potentially" or another modifier before "capable of producing..."

Answer: We thank the reviewer for the comments and suggestions. We changed the text in the revised manuscript (page 7, line 139).

Comment 2: Figure 1 j,k and n, and Page 15 lines 348-9. Another reviewer brought up this issue and it has not been resolved. I believe the y-axis is incorrect and should be 10^4 not 10^{-4} (j and k) and 3 for n. On page 15 there is also confusion as to total cell number.

Answer: We apologize for this minor oversight. We changed the figure 1 and the text (page 16, line 366-367) in the revised manuscript.

Reviewer #3 (Remarks to the Author):

The authors have made reasonable efforts to address the issues that I and the other reviewers made on the original version of the manuscript, and the manuscript is substantially improved.

Areas which I believe remain somewhat unclear include:

Comment 1: How critical the deficiency in pancreatic enzyme production is to the phenotype (wasting and death). The authors claim it is not feasible to address this as gavage is not feasible in neonatal mice.

Answer: We cannot categorically say that the pancreatic deficit is the only factor contributing to wasting and death of the animals. We point out in the manuscript (page 11-12, line 252-267, figure 6) that absorption pathways are also affected at the level of the small intestine. What we can say is that IL-22 is the main factor controlling both pancreatic and intestinal deficits as they are normalized in the absence of IL-22. Transgenic animals expressing IL-23 but lacking IL-22 survive and grow normally.

2. Whether the alterations in the exocrine pancreas reflect differentiation changes in cells evolving from stem cells rather than de-differentiation of acinar cells. The data, including some new data, seems to favour the former and may be an effect of IL-22 only applicable to neonates.

Having highlighted these issues, the manuscript has some caveats around these two points now so that is more acceptable.

Prof Michael McGuckin

Answer: We have adjusted the discussion according to reviewer's suggestion in the revised manuscript (page 18, line 415-420 in our revised manuscript).

Reviewer #4 (Remarks to the Author):

The authors largely addressed by comments in the revised manuscript.

Answer: Many thanks.

REVIEWERS' COMMENTS:

Reviewer #2 (Remarks to the Author):

The authors have addressed my comments from the last review. The new data on necrotizing enterocolitis adds to the manuscript and is well presented and proper conclusions are presented.

Reviewer #2 (Remarks to the Author):

The authors have addressed my comments from the last review. The new data on necrotizing enterocolitis adds to the manuscript and is well presented and proper conclusions are presented.

Answer: Many thanks.